# PRECISE ATTRIBUTE INTENSITY CONTROL IN LARGE LANGUAGE MODELS VIA TARGETED REPRESENTATION EDITING

## ABSTRACT

Precise attribute intensity control—generating Large Language Model (LLM) outputs with specific, user-defined attribute intensities—is crucial for AI systems adaptable to diverse user expectations. Current LLM alignment methods, however, typically provide only directional or open-ended guidance, failing to reliably achieve exact attribute intensities. We address this limitation with three key designs: (1) reformulating precise attribute intensity control as a target-reaching problem, rather than simple maximization; (2) training a lightweight value function via temporal-difference learning to predict final attribute intensity scores from partial generations, thereby steering LLM outputs; and (3) employing gradient-based interventions on hidden representations to navigate the model precisely towards specific attribute intensity targets. Our method enables fine-grained, continuous control over attribute intensities, moving beyond simple directional alignment. Experiments on LLaMA-3.2-3b and Phi-4-mini confirm our method's ability to steer text generation to user-specified attribute intensities with high accuracy. Finally, we demonstrate efficiency enhancements across three downstream tasks: preference data synthesis, Pareto frontier approximation and optimization, and distillation of aligned behaviors for intervention-free inference. Our code is available on https://anonymous.4open.science/r/pre-control-F482.

## 1 INTRODUCTION

Precise control over attribute intensity is critical for tailoring large language model (LLM) outputs to diverse contexts and user needs [2, 32]. Rather than merely pushing attributes in a single direction, precise attribute intensity control enables fine-grained adjustment of text attributes—such as tone, helpfulness, or formality—on a continuous scale [8, 35]. This capability is essential for practical applications, such as calibrating an email's tone from slightly formal for a colleague to highly formal for an executive [7]. The stakes are even higher in multi-objective alignment, where attributes conflict with each other [4, 50]. Navigating trade-offs between attributes, such as maximizing helpfulness while minimizing misinformation, requires scalar-level adjustments to identify optimal compromises [2, 47]. However, adjusting an LLM along continuous attribute trade-offs is difficult. While sophisticated prompting can elicit complex behaviors, it remains an unreliable method for precise and reproducible attribute control. The mapping from a qualitative description to a point in the model's attribute space is non-trivial and highly sensitive to phrasing. This indirect mechanism makes it challenging to achieve specific scalar targets, especially when attributes are entangled in multi-objective scenarios [11, 37].

Existing alignment paradigms fundamentally lack the capability for efficient precise attribute intensity control. Fine-tuning methods like Reinforcement Learning from Human Feedback (RLHF; [37, 53, 40]) and direct preference optimization (DPO; [35]) produce static models that capture an average of desired behaviors, requiring expensive retraining to shift priorities. While recent advances in multi-objective alignment [36, 52, 48, 51] can identify Pareto-optimal solutions, they often require extensive training to approximate a global Pareto set rather than enabling efficient, controllable projection of individual generations onto specific points on that frontier. Test-time methods avoid retraining but have their own limitations. Prompting approaches [1, 49, 21] rely entirely on the model's interpretation of style instructions, yielding inconsistent

results. Guided decoding [16, 29, 14, 15] typically treat attribute intensity as categorical rather than continuous. Moreover, without modifying the model's parameters, these methods remain constrained by the pretrained model's capabilities, making effective fine-grained control (*e.g.*, adjusting helpfulness$= 4$, complexity$= 2$ on a $0 - 4$ scale) unattainable.

We address this gap by introducing a method for precise control over attribute intensity via targeted representation editing. Our method, named PRE-CONTROL, consists of three key innovations: (1) To enable users to specify target values for preference attributes, we formulate precise attribute intensity control as a target-reaching problem rather than merely maximizing or minimizing values. This shift is necessary because achieving specific attribute intensities requires optimization toward exact target values rather than extremal points. (2) To provide guidance during the generation process, we train a lightweight value function using temporal-difference learning. The value function predicts final attribute scores from partial generations, which significantly improves efficiency by allowing real-time adjustments during LLM decoding rather than requiring multiple complete generations and post-hoc evaluations to achieve target attribute intensity. (3) To precisely navigate the high-dimensional representation space toward specific attribute targets, we employ gradient-based interventions on the hidden representation space of LLMs. Together, these components enable PRE-CONTROL to offer finer granularity in aligning LLM behavior, producing outputs that match concrete attribute specifications rather than vaguely "more aligned" responses.

Experiments on multi-objective preference datasets using LLaMA-3.2-3b and Phi-4-mini demonstrate significantly higher success rates, in achieving user-specified target attribute scores compared to baseline methods. This capability enables two downstream applications. (1) *Efficient Pareto frontier approximation*. Traditional methods for approximating Pareto frontiers require exhaustive sampling across preference attributes combinations (scaling poorly as $O(m^d)$ for $m$ attributes and $d$ dimensions). In contrast, PRE-CONTROL dramatically reduces the time complexity to $O(n+k)$ while maintaining frontier quality, making multi-objective preference optimization practical for high-dimensional attribute spaces. (2) *Controllable model distillation*. We leverage PRE-CONTROL to efficiently generate training data with specific attribute intensity. Unlike conventional approaches that rely on best-of-N sampling or random sampling with filtering, our method directly generates examples at any target attribute intensity, creating comprehensive training datasets that enable models to learn aligned behaviors for intervention-free inference.

## 2 PRELIMINARIES

### 2.1 FROM STANDARD LLM ALIGNMENT TO TARGET REACHING FORMULATION

We formalize the problem of precise attribute intensity control in LLMs by contrasting it with standard alignment objectives. Let $\pi_\theta(x_t|x_{<t})$ be a language model parameterized by $\theta$, which generates tokens $x_t$ conditioned on the history $x_{<t}$. Traditional alignment approaches aim to improve the model's outputs according to human preferences, typically represented by a preference or reward function $R(x) \in \mathbb{R}$ that evaluates how well a text sequence $x$ exhibits a desired attribute. In conventional alignment frameworks such as RLHF [32], the objective is typically formulated as:

$$\max_\theta \mathbb{E}_{x \sim \pi_\theta}[R(x)], \quad (1)$$

which aims to find parameters $\theta$ that maximize the expected reward across generated sequences. This approach focuses on pushing the model outputs in a single direction—toward higher reward values.

We propose a shift from "*optimizing for the maximum (or minimum) reward values*" to "*reaching a specific target attribute intensity*". Let $\tau \in [0, 1]$ denote a normalized target attribute intensity score specified by the user. Given a reward function $R(x)$ with range $[R_{min}, R_{max}]$, we define a normalized reward function $\hat{R}(x) = \frac{R(x) - R_{min}}{R_{max} - R_{min}}$, such that $\hat{R}(x) \in [0, 1]$. Our objective then becomes:

$$\min_\theta \mathbb{E}_{x \sim \pi_\theta}[(\hat{R}(x) - \tau)^2]. \quad (2)$$

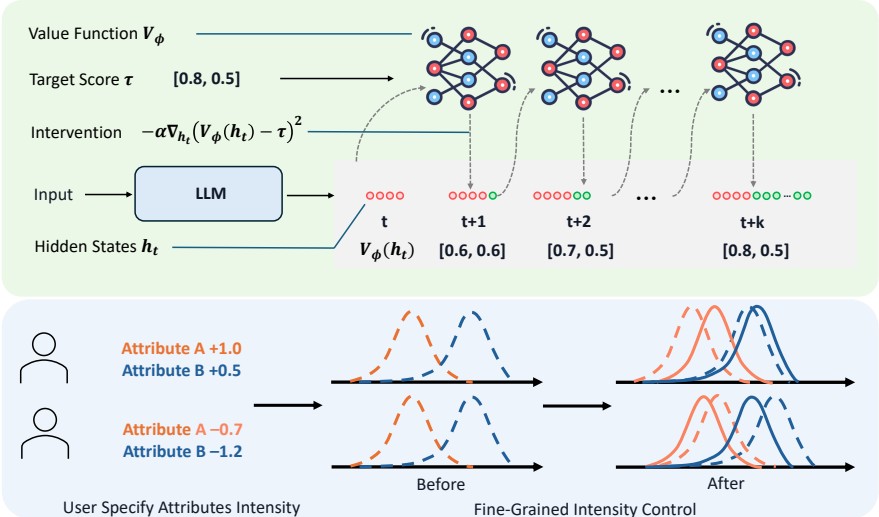

Figure 1: Overview of PRE-CONTROL. For precise attribute intensity control, we formalize it as a target-reaching problem. We train a value function on the hidden space of an LLM to predict the attribute-wise reward. During test-time, we leverage this value function to guide the LLM generating text towards the specified attribute scores through targeted representation editing.

This formulation explicitly aims to generate text whose attribute intensity score matches the target value $\tau$, rather than simply maximizing or minimizing the preference. The squared error term penalizes deviations from the target in either direction, enabling precise control over the strength of the attribute.

## 2.2 PRECISE MULTI-ATTRIBUTE INTENSITY CONTROL

Real-world applications often require balancing multiple attributes simultaneously. Let $\mathbf{R} = R_1, R_2, ..., R_m$ be a set of $m$ reward functions corresponding to different attributes (e.g., helpfulness, safety, complexity), and $\boldsymbol{\tau} = \tau_1, \tau_2, ..., \tau_m$ their target levels. The multi-attribute target-reaching problem can be formulated as:

$$\min_{\theta} \mathbb{E}_{x \sim \pi_\theta} \left[ \sum_{i=1}^{m} w_i (\hat{R}_i(x) - \tau_i)^2 \right], \tag{3}$$

where $w_i \geq 0$ weight the relative importance of each attribute. This formulation allows for nuanced control across multiple dimensions of model behavior simultaneously, where each attribute can be tuned to a specific level rather than simply maximized or minimized. For instance, a user might want to set helpfulness to a very high level ($\tau_{\text{helpfulness}} = 0.9$) while maintaining only moderate complexity ($\tau_{\text{complexity}} = 0.5$).

## 3 PRECISE ATTRIBUTE INTENSITY CONTROL VIA TARGET REPRESENTATION EDITING

In this section, we present our method for precise attribute intensity control that enables language models to generate outputs with user-specified attribute intensity. Our approach consists of two core components: (1) value function training that predicts expected attribute intensity scores from partial generations, and (2) test-time intervention that guides the generation process toward target attribute intensity. We also demonstrate an efficient technique for Pareto frontier approximation as a practical application.

### 3.1 VALUE FUNCTION TRAINING VIA TEMPORAL DIFFERENCE LEARNING

The key challenge in precise attribute intensity control for LLM is providing accurate guidance during decoding. Traditional methods only evaluate complete sequences, offering no intermediate feedback that could guide partial generations toward desired attribute intensity. To address this limitation, we train a value function that predicts the expected attribute intensity of a complete generation based on partial sequences. Given a model $\pi_\theta(x_t|x_{<t})$ that generates tokens $x_t$ conditioned on history $x_{<t}$, we define a value function $V_\phi(h_t)$ that maps from the model's hidden state $h_t$ at decoding step $t$ to a predicted attribute intensity:

$$V_\phi(h_t) \approx \mathbb{E}_{x_{>t} \sim \pi_\theta(\cdot|x_{\leq t})} \left[ \hat{R}(x_{\leq t}, x_{>t}) \right]. \tag{4}$$

Here, $\hat{R}$ represents the normalized reward function mapping to $[0, 1]$ as defined in Section 2.1. Training such a value function $V_\phi(h_t)$ through supervised learning would require expensive rollouts to obtain ground truth labels for each partial sequence. Instead, we adopt TD($\lambda$) [39], a temporal-difference method that enables the value function to efficiently learn by bootstrapping from future predictions. We compute a generalized return incorporating multiple future milestone rewards:

$$G_t^\lambda = (1 - \lambda) \sum_{n=1}^{T-t-1} \lambda^{n-1} V_\phi(s_{t+n}) + \lambda^{T-t-1} r^T \tag{5}$$

In this formulation, $s_{t+n}$ denotes the state reached $n$ steps after $s_t$ (in our setting, $s_t := h_t$), where $T$ represents the total sequence length. The term $V_\phi(s_{t+n})$ serves as a bootstrap estimate of the eventual terminal score starting from that future state, while $r^T$ is the final, episode-level reward for the completed sequence. The decay factor $\lambda$ trades off short-horizon bootstrapping against reliance on the terminal Monte Carlo target—approaching pure MC as $\lambda \to 1$. The value function is then trained to minimize the mean squared error between its predictions and the generalized returns:

$$\mathcal{L}_{TD} = \mathbb{E}_{t,s_t} \left[ (V_\phi(s_t) - G_t^\lambda)^2 \right]. \tag{6}$$

This TD($\lambda$) approach provides crucial intermediate feedback signals that were previously missing in preference alignment methods [18]. The decay factor enables proper credit assignment across time steps, allowing the value function to provide reliable guidance at each generation step.

In practice, we implement the value function as a multi-layer perceptron (MLP) that operates on the hidden representations of LLMs. The value function is trained on a diverse corpus of pre-generated texts annotated with attribute intensity scores from an external reward model, simulating the generation process and computing generalized returns at multiple timesteps.

### 3.2 TEST-TIME INTERVENTION FOR TARGET ATTRIBUTE INTENSITY CONTROL

With a trained value function that can predict attribute intensity scores from partial generations, we leverage it to guide the language model toward generating text with a specific attribute score through targeted representation editing. Unlike previous approaches [15, 18, 21, 49] that merely push the model to maximize or minimize a preference, our method enables precise targeting of any scores within the full range of attribute intensities.

Given a target attribute intensity score $\tau \in [0, 1]$, we aim to minimize the deviation between the predicted attribute intensity score and the target:

$$\min_{h_t} \left( V_\phi(h_t) - \tau \right)^2. \tag{7}$$

We achieve this through gradient descent on the hidden states during the generation process. At each decoding step $t$, we compute the prediction of the value function $V_\phi(h_t)$ based on the current hidden state $h_t$. If the predicted score deviates from the target $\tau$, we adjust the hidden state through:

$$h_t \leftarrow h_t - \alpha \nabla_{h_t} \left( V_\phi(h_t) - \tau \right)^2. \tag{8}$$

The step size $\alpha$ controls the strength of the intervention. This gradient-based adjustment steers the hidden state toward a region that is expected to lead to a generation with the target attribute intensity score. The intervention minimizes the deviation between the predicted attribute intensity score and the target score, enabling controlled and fine-grained adjustment that ensures outputs align precisely with the desired preference strength.

For scenarios requiring control over multiple preference attributes simultaneously, our value function $V_\phi$ outputs a vector of attribute intensity scores $[V_\phi^1(h_t), V_\phi^2(h_t), ..., V_\phi^m(h_t)]$, where each element corresponds to a different preference attribute. Given a vector of target attribute intensity scores $\boldsymbol{\tau} = [\tau_1, \tau_2, ..., \tau_m]$, we extend our gradient descent approach to minimize the weighted deviation across all attributes:

$$h_t \leftarrow h_t - \alpha \nabla_{h_t} \sum_{i=1}^{m} w_i (V_\phi^i(h_t) - \tau_i)^2, \tag{9}$$

where $w_i$ represents the weight determining the relative importance of each attribute.

This formulation enables fine-grained control across multiple dimensions of text quality simultaneously. Our test-time intervention approach offers several advantages over existing methods. Unlike prompting or RLHF, which push models toward binary or categorical outcomes, our method enables continuous, fine-grained control over preference strength. The value function provides real-time feedback during generation, allowing for adaptive adjustments based on the current state. Additionally, our method works with existing pre-trained models without requiring expensive fine-tuning for each target attribute intensity. By making minimal, targeted interventions, we maintain the model's underlying knowledge and capabilities while adjusting only the preference-related aspects.

### 3.3 EFFICIENT PARETO FRONTIER APPROXIMATION

**Phase 2: Interpolation Target Generation.**   To explore the gaps in our initial frontier approximation, we generate a set of target points $\mathcal{T}$ by interpolating between adjacent non-dominated points. For each pair of adjacent points $(\mathbf{n}_1, \mathbf{n}_2) \in \mathcal{N}$, we generate $K$ interpolated points using an interpolation function $I$:

$$\mathbf{t} = I(\mathbf{n}_1, \mathbf{n}_2, \beta), \quad \beta \in [0, 1] \tag{11}$$

where $\beta$ is the interpolation coefficient that controls how close the target point $\mathbf{t}$ is to each of the non-dominated points. While simple linear interpolation is often sufficient, our method is compatible with arbitrary interpolation strategies.

**Phase 3: Targeted Refinement.**   The core of our approach is using our precise attribute intensity control capability to directly generate samples at specific target points along the Pareto frontier, which traditional methods cannot achieve. For each iteration, we identify the most promising target by calculating the coverage gap at each point $\mathbf{t}$ as:

$$G(\mathbf{t}, \mathcal{N}) = \min_{\mathbf{n} \in \mathcal{N}} |\mathbf{t} - \mathbf{n}|_2. \tag{12}$$

We select the target point $\mathbf{t}^*$ with the largest coverage gap and apply our test-time intervention to guide the language model toward generating a sample with attribute intensity scores matching this multi-dimensional target. By precisely controlling the generation process to reach specific combinations of preference attributes, we can efficiently discover new non-dominated points in underexplored regions.

**Efficiency Advantage.** By leveraging precise attribute intensity control, our method significantly improves Pareto frontier approximation efficiency. Traditional approaches either require grid sampling across preference weights (scaling as $O(m^d)$ for $d$ dimensions) or training separate models for different preference combinations. In contrast, PRE-CONTROL identifies non-dominated points from initial samples, interpolates between them to generate promising targets, and uses value function-guided intervention to steer generation precisely toward these targets. This targeted exploration achieves comparable frontier coverage while requiring much fewer computation costs ($O(n + k)$ where $n$ is the number of initial samples and $k$ is the refinement budget). compared to baseline methods. We evaluate these computational advantages in Section 4.4.

An important application of PRE-CONTROL is efficiently approximating the Pareto frontier for multiple competing preference attributes. Given $m$ preference attributes with scores $\mathbf{R} = [R_1, R_2, ..., R_m]$, the Pareto frontier $\mathcal{P}$ is defined as the set of all non-dominated points in the attribute intensity space. Formally, a point $\mathbf{p} \in \mathcal{P}$ if and only if there does not exist another achievable point $\mathbf{q}$ such that:

$$\forall i \in \{1, 2, ..., m\}: \quad R_i(\mathbf{q}) \geq R_i(\mathbf{p})$$
$$\text{and} \quad \exists j \in \{1, 2, ..., m\}: \quad R_j(\mathbf{q}) > R_j(\mathbf{p}). \tag{10}$$

Approximating Pareto frontier is typically computationally expensive, requiring exhaustive sampling or training separate models. To this end, we populate the frontier by conditioning each generation on a distinct target attribute vector located along the trade-off surface. We propose Algorithm 1 that leverages our precise attribute intensity control capabilities to systematically explore the preference space with significantly fewer model calls. This algorithm consists of three phases:

**Phase 1: Initial Sampling.** We first generate a set of samples $\mathcal{S}$ from the base language model and evaluate them on all preference attributes. From these samples, we extract the set of non-dominated points $\mathcal{N}$ to form our initial approximation of the Pareto frontier.

---

**Algorithm 1** Efficient Pareto Frontier Approximation

---

**Require:** Model $\pi_\theta$, value function $V_\phi$, interpolation function $I$
**Ensure:** Approximated Pareto frontier $\mathcal{P}$
1: **Phase 1: Initial Sampling**
2: $\mathcal{S} \leftarrow$ Generate base samples from $\pi_\theta$
3: Evaluate all samples on preference attributes
4: $\mathcal{N} \leftarrow$ Extract non-dominated points from $\mathcal{S}$
5: **Phase 2: Interpolation Target Generation**
6: $\mathcal{T} \leftarrow \emptyset$
7: **for** each adjacent pair $(\mathbf{n}_1, \mathbf{n}_2) \in \mathcal{N}$ **do**
8:     **for** $k = 1$ to $K$ **do**
9:       $\lambda \leftarrow \frac{k}{K+1}$
10:       $\mathbf{t} \leftarrow I(\mathbf{n}_1, \mathbf{n}_2, \lambda)$
11:       $\mathcal{T} \leftarrow \mathcal{T} \cup \{\mathbf{t}\}$
12:     **end for**
13: **end for**
14: **Phase 3: Targeted Refinement**
15: **while** refinement budget not exhausted **do**
16:     $\mathbf{t}^* \leftarrow \arg\max_{\mathbf{t} \in \mathcal{T}} G(\mathbf{t}, \mathcal{N})$
17:     Generate sample from $\pi_\theta$ with intervention toward $\mathbf{t}^*$
18:     Update $\mathcal{N}$ with new non-dominated points
19:     $\mathcal{T} \leftarrow \mathcal{T} \setminus \{\mathbf{t}^*\}$
20: **end while**
21: $\mathcal{P} \leftarrow \mathcal{N}$
22: **return** $\mathcal{P}$

---

## 4 EXPERIMENT

### 4.1 EXPERIMENTAL SETUP

**Dataset.** We conduct experiments on HelpSteer2 [43] and Code-UltraFeedback [44], two multi-attribute datasets for LLM alignment. HelpSteer2 (20k samples) and Code-UltraFeedback (10k samples) are annotated with Likert-scale scores (0–4) on five attributes. The attributes span general dialogue quality—*helpfulness*, *correctness*, *coherence*, *complexity*, and *verbosity*—in HelpSteer2, and code-specific aspects—*complexity and efficiency*, *style*, *explanation*, *instruction-following*, and *readability*—in Code-UltraFeedback. These structured annotations support fine-grained supervision and evaluation of attribute intensity control in multi-objective settings, where trade-offs between conflicting attributes are often required [31].

**Models.** We evaluate our method using two base models: LLaMA-3.2-3b [12] and Phi-4-mini [28]. For the value function, we train a 4-layer MLP that takes hidden representations from the base models as input to predict their corresponding (normalized) reward scores. The supervision signals are provided by a publicly available reward model ArmoRM[1] [42], which is externally trained to predict multi-attribute attribute intensity scores. We extract hidden representations from the final layer of each base model and apply intervention at this layer. This design choice is motivated by prior work [9, 23], which shows that upper layers in transformer

---

[1]https://huggingface.co/RLHFlow/ArmoRM-Llama3-8B-v0.1

models encode more semantic and task-specific information, making them suitable for reward estimation and intervention. In addition, intervening only at the final layer reduces interference with lower-level features and offers a more efficient control mechanism. We find that this implementation achieves strong empirical performance, and we leave the exploration of multi-layer or attention-level intervention to future work.

**Metrics.** Following [10], we leverage **Self-BLEU** score to measure the diversity of generated text. A lower Self-BLEU score suggests higher textual diversity. $\ell_1$ **Distance to Target** evaluates how closely the model output aligns with the user-specified attribute scores. Each target is a 5-dimensional vector, representing desired scores across five attributes. Lower values indicate better precision in attribute intensity control. **Success Rate** quantifies how often the model output exactly matches the desired attribute configuration. It is calculated as $\frac{N_{\text{Aligned samples after intervention}}}{N_{\text{Misaligned samples before intervention}}}$. We use ArmoRM to score the generated responses and round the score into its nearest integer. The intervention is considered as success if the rounded score matches the target score exactly in all dimentions. To ensure meaningful evaluation, we filter out samples whose base model responses already align with the target reward and apply PRE-CONTROL on those unsatisfied samples.

**Baselines.** We compare our method with the following methods. **Base:** The base model directly generates responses without any explicit control over attributes intensity. **Prompting:** Prompting steers model outputs by incorporating target attribute scores directly into the prompt. We follow the prompting practice of [12], where the instruction includes the scale description and desired attribute values. **Static Representation:** ITI [19] trains a linear layer to predict reward from LLM hidden states, then shifts activations along the learned direction using a fixed vector throughout generation. **Multi-attribute Steering:** MAT-Steer [31] learns sparse, orthogonal steering vectors for multiple attributes to reduce inter-attribute conflicts. **Representation Editing:** RE-Control [18] performs test-time intervention, which is an open-ended optimization procedure that pushes the hidden representations in a monotonic direction.

## 4.2 MAIN RESULTS

We evaluate the effectiveness of PRE-CONTROL for precise attribute intensity control on HelpSteer2 and Code-UltraFeedback. Table 1 presents the main results on both relative positive and negative target vectors, which illustrate PRE-CONTROL's bidirectional finer-grained control capability. Crucially, the strong performance of our method is not limited to these specific points. We provide a comprehensive evaluation across a wide range of target scores in Appendix C.3, with full results in Table 14 and 15, which confirms the robustness and consistency of our findings.

We summarize our findings as follows: **(1) Superior Success Rate and Improvement Margin.** PRE-CONTROL consistently achieves the highest success rates across all settings, where the success rates ranges from 6.60% to 30.68%, representing improvements of up to 5.1× over the best baseline. This bidirectional capability – equally effective at both increasing and decreasing attribute intensities – is crucial for multi-objective alignment, as navigating trade-offs between competing attributes is essential when optimizing for Pareto-optimal solutions. Unlike methods that can only maximize preferences, our approach enables precise targeting of any point within the attribute space, making it particularly valuable for applications requiring nuanced control over multiple objectives simultaneously. **(2) Enhanced Diversity with Maintained Quality.** Using Self-BLEU as our diversity metric, PRE-CONTROL achieves the lowest scores across all conditions – as low as 0.291 for HelpSteer2 and 0.279 for Code-UltraFeedback – indicating significantly more diverse outputs compared to baselines. This diversity suggests that our method avoids the mode collapse often seen in traditional alignment approaches, while still maintaining precise control over attribute intensities. **(3) Consistent Performance Across Models and Datasets.** Our method demonstrates robust performance improvements on both LLaMA-3.2-3b and Phi-4-mini across two distinct domains. Additional experiments on Phi-4-mini show consistent improvements: 26.16% success rate (vs. 18.92% for MAT-Steer) on positive targets and 22.34% (vs. 8.38%) on negative targets. This generalizability suggests that our value function learning and intervention approach works well across different model architectures and task types.

| | | HelpSteer2 [4, 4, 4, 2, 2] | | | Code-UltraFeedback [3, 3, 3, 3, 3] | | |
|---|---|---|---|---|---|---|---|
| **Dataset and Target Score** | | | | | | | |
| **Backbone** | **Method** | **Diversity ↓** | **$\ell_1$ Distance to Target ↓** | **Success Rate (%) ↑** | **Diversity ↓** | **$\ell_1$ Distance to Target ↓** | **Success Rate (%) ↑** |
| Llama-3.2-3B | Base | 0.626 | 2.19 | N/A | 0.876 | 2.29 | N/A |
| | Prompting | 0.941 | 2.17 | 5.39 | 0.879 | 2.21 | 6.80 |
| | ITI | 0.604 | 3.02 | 3.75 | 0.741 | 2.62 | 12.72 |
| | Re-Control | 0.946 | 2.16 | 5.39 | 0.880 | 2.21 | 7.54 |
| | MAT-Steer | 0.739 | 2.22 | 5.17 | 0.778 | 2.41 | 13.63 |
| | **Ours** | **0.558** | **2.16** | 7.96 | **0.614** | **2.08** | **17.46** |
| Phi-4-mini (3.8B) | Base | 0.701 | 2.46 | N/A | 0.902 | 1.57 | N/A |
| | Prompting | 0.698 | 2.42 | 5.23 | 0.903 | 1.47 | 9.46 |
| | ITI | 0.534 | 3.63 | 2.61 | 0.789 | 1.55 | 16.49 |
| | Re-Control | 0.611 | 2.51 | 5.70 | 0.786 | 1.43 | 17.25 |
| | MAT-Steer | 0.503 | 2.46 | 5.48 | 0.700 | 1.43 | 18.92 |
| | **Ours** | **0.530** | **2.41** | **8.31** | **0.688** | **1.42** | **26.16** |
| Phi-4 (14B) | Base | 0.918 | 2.04 | N/A | 0.979 | 0.694 | N/A |
| | Prompting | 0.946 | 2.01 | 1.93 | 0.987 | 0.632 | 10.34 |
| | ITI | 0.709 | 2.04 | 5.06 | 0.920 | 0.584 | 16.33 |
| | Re-Control | 0.914 | 1.99 | 3.61 | 0.972 | 0.656 | 11.29 |
| | MAT-Steer | 0.697 | 2.29 | 3.86 | 0.912 | 0.569 | 21.66 |
| | **Ours** | **0.686** | **1.93** | **7.23** | **0.880** | **0.560** | **26.96** |

**Relative Negative Representative Target Score**

| | | HelpSteer2 [3, 3, 3, 2, 2] | | | Code-UltraFeedback [2, 2, 2, 2, 2] | | |
|---|---|---|---|---|---|---|---|
| **Dataset** | | | | | | | |
| **Backbone** | **Method** | **Diversity ↓** | **$\ell_1$ Distance to Target ↓** | **Success Rate (%) ↑** | **Diversity ↓** | **$\ell_1$ Distance to Target ↓** | **Success Rate (%) ↑** |
| Llama-3.2-3B | Base | 0.656 | 2.76 | N/A | 0.874 | 2.95 | N/A |
| | Prompting | 0.987 | 2.73 | 2.47 | 0.865 | 2.85 | 6.06 |
| | ITI | **0.294** | 2.69 | 5.48 | 0.441 | 2.83 | 6.79 |
| | Re-Control | 0.986 | 2.72 | 2.27 | 0.607 | 2.78 | 6.57 |
| | MAT-Steer | 0.539 | **2.57** | 5.84 | 0.480 | 2.59 | 16.67 |
| | **Ours** | 0.251 | 2.63 | 6.60 | **0.440** | **1.95** | **30.68** |
| Phi-4-mini (3.8B) | Base | 0.659 | 2.76 | N/A | 0.868 | 3.65 | N/A |
| | Prompting | 0.664 | 2.67 | 5.18 | 0.869 | 3.64 | 2.15 |
| | ITI | 0.450 | 2.73 | 4.02 | 0.623 | 3.66 | 4.54 |
| | Re-Control | 0.494 | 2.56 | 5.80 | 0.614 | 3.53 | 6.92 |
| | MAT-Steer | 0.308 | 2.86 | 8.73 | 0.318 | 2.89 | 8.38 |
| | **Ours** | **0.291** | **2.46** | **9.11** | **0.279** | **2.80** | **22.34** |
| Phi-4 (14B) | Base | 0.930 | 2.79 | N/A | 0.963 | 4.58 | N/A |
| | Prompting | 0.962 | **2.78** | 1.25 | 0.974 | 4.57 | 0.31 |
| | ITI | 0.656 | 2.83 | 2.02 | 0.778 | 4.62 | 1.65 |
| | Re-Control | 0.888 | 2.82 | 2.49 | 0.917 | 4.56 | 0.72 |
| | MAT-Steer | 0.644 | 2.72 | 2.73 | 0.651 | 4.59 | 1.65 |
| | **Ours** | **0.602** | 2.83 | **3.74** | **0.605** | 2.89 | **19.14** |

Table 1: Main results on representative target scores. These targets are defined based on the statistical distribution of attributes combination in each dataset (detailed in Figure 5). These targets serve as illustrative examples, Appendix C.3 presents a comprehensive evaluation across a wider range of target scores.

### 4.3 ITERATIVE RESULTS OF ATTRIBUTE INTENSITY CONTROL

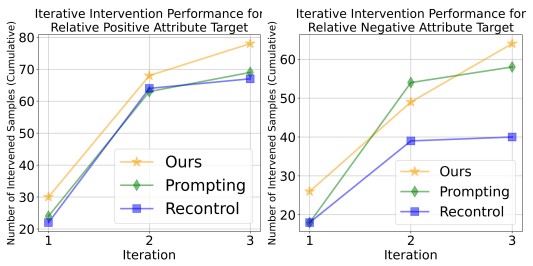

Figure 2: Iterative intervention results.

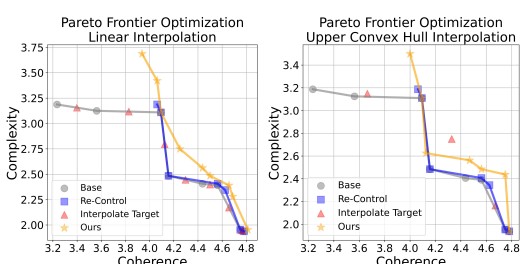

Figure 3: Pareto frontier comparison.

Figure 2 shows the cumulative performance across multiple intervention iterations on HelpSteer2. In order to continuously steer the generation towards the desired attribute intensity, we append the model's response from the last intervention iteration to prompt and ask it to re-address the question. We have the following

### 4.4 Pareto Frontier Approximation

In this set of experiments, we leverage Pre-Control to approximate Pareto frontier and study its quality and efficiency. We choose a pair of conflicting preference attributes (*coherence vs. complexity*) from HelpSteer2 and follow the procedure in Algorithm 1 to obtain the initial Pareto frontier from the base model and the improved Pareto frontiers with the studied methods.

Figure 3 demonstrates that Pre-Control establishes a more dominant Pareto frontier compared to both RE-Control and the base model. This is evident across both linear and upper convex hull interpolation strategies, showing that our method consistently achieves better trade-offs among conflicting attributes.

Figure 4 further plots the attribute-wise reward distributions for coherence and complexity, contrasting the reward scores before and after the application of Pre-Control. After the intervention, both distributions shift towards higher reward scores. This simultaneous positive movement in both Coherence and Complexity rewards is significant, indicating our method's ability to enhance outputs across multiple attributes concurrently. Such improvements suggest our approach can effectively guide the LLM to cover more dominant regions of the Pareto frontier.

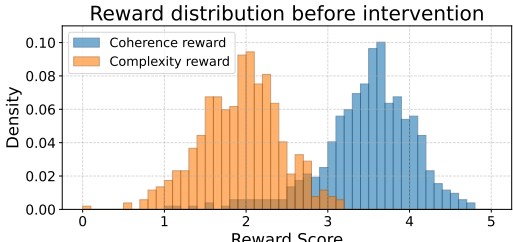

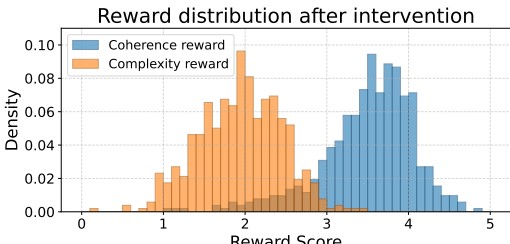

Figure 4: Attribute-wise reward distribution.

| Method | HV | Sparsity | # PF | Overhead |
|--------|------|----------|------|----------|
| Base | 7.03 | 0.41 | 29 | – |
| GS | 7.54 | **0.21** | 45 | 3.3 |
| **Ours** | **12.66** | 0.24 | **69** | **0.4** |

Table 2: Pareto frontier approximation quality and efficiency. Overhead is measured in GPU hours.

observations. First, Pre-Control consistently exhibits the best cumulative performance for both positive and negative targets. It establishes an early lead that significantly widens by the third iteration (*e.g.*, reaching nearly 80 intervened samples for positive and approximately 65 for negative targets), highlighting its strong benefit from iterative refinement over other methods.

Second, Prompting displays a notable performance surge in its second iteration, particularly for negative targets where its cumulative intervened samples jump from approximately 28 to over 50. This second-round boost is attributable to its design of using previous responses as in-context demonstrations. Nevertheless, Prompting's final performance remains below that of Pre-Control, emphasizing the robustness of our representation editing method.

Third, both Prompting and Recontrol plateau after the second iteration. Prompting is limited by its heavy dependency on the model's interpretation of style-based instructions, a process that can yield inconsistent outputs and thus impede steady, cumulative refinement. REControl is limited by its open-ended control, which struggles to precisely steer the model towards a specified target intensity. In summary, these methods lack an effective mechanism for consistent and targeted adjustment of attribute intensity across multiple iterations, unlike the progressive improvements observed with Pre-Control.

Table 2 quantifies more Pareto frontier metrics – HV refers to the hypervolume, which measures the dominated space volume; sparsity measures the average distance between adjacent non-dominated points (lower is better); and #PF indicates the number of non-dominated points discovered. We highlight the efficiency of Pre-Control, which achieves substantially higher hypervolume (12.66 vs. 7.54 for grid sampling (GS)) and discovers more Pareto-optimal points (69 vs. 45) while requiring only 0.4 GPU hours compared to GS's

3.3 hours. This 8× reduction in computational overhead demonstrates that our approach not only produces higher-quality Pareto frontier approximations but does so with significantly greater efficiency.

### 4.5 ADDITIONAL EXPERIMENTS

We further evaluate PRE-CONTROL in complementary experiments: (i) *iterative intervention* (Appendix C.2); (ii) *iterative Pareto-frontier approximation* (Appendix C.4); and (iii) *controllable distillation* (Appendix C.5). Full results are provided in the corresponding appendices.

## 5 CONCLUSION

We presented PRE-CONTROL, a framework for precise attribute intensity control in LLMs via targeted representation editing. By reformulating alignment as a target-reaching problem, we enable fine-grained control over preference attributes on a continuous scale through value function learning and gradient-based hidden state interventions. Experiments on LLaMA-3.2-3b and Phi-4-mini demonstrate that PRE-CONTROL significantly outperforms baselines in achieving user-specified attribute intensities while maintaining text quality. Our approach enables Pareto frontier approximation with reduced computational complexity, and efficient controllable model distillation using 3.3× fewer samples than best-of-N approaches. We further discuss limitations and future research directions in Appendix A.

## 6 REPRODUCIBILITY STATEMENT

We release code, configs, and scripts at https://anonymous.4open.science/r/pre-control-F482. The core algorithmic details are specified in Section 3 and Algorithm 1, and the full experimental setup appears in Section 4. Dataset descriptions and preprocessing steps for HelpSteer2 and Code-UltraFeedback are provided in Appendix D.1 and Appendix D.2, respectively. Implementation specifics—including model/backbone choices, intervention layer, value-function architecture, and training targets (ArmoRM), hyperparameters, random seeds, and decoding settings—are documented in Appendix D.3 and in the released configuration files. Our Pareto-frontier construction and interpolation procedures, along with the hypervolume, sparsity, and #PF computations, are detailed in Appendix D.4; metric definitions (Self-BLEU, $\ell_1$ distance to target, and Success Rate with filtering rules) are summarized in the Metrics subsection of Section 4 and mirrored in the repository's evaluation scripts. Computing infrastructure (hardware, GPU hours, and environment) is reported in Appendix E.

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

# Appendix for PRE-CONTROL

## A   LIMITATIONS AND FUTURE WORK

**Value Function as Reward Model Proxy.** To pursue efficiency in value function training and intervention efficiency, we employ a lightweight MLP as a value function that learns from reward model outputs. While this design choice enables efficient real-time intervention, it inherently sacrifices some accuracy compared to directly using the full reward model. The value function serves as a proxy that may not capture all nuances of the original reward signal. Future work could explore more sophisticated architectures that better approximate reward model capabilities while maintaining computational efficiency, or investigate adaptive mechanisms that selectively query the full reward model for challenging cases.

**Final Layer Intervention.** Our current implementation applies interventions at the final transformer layer. While this design choice yields strong empirical results and computational efficiency, it may not fully exploit the model's representation hierarchy. Future research could explore multi-layer intervention strategies or develop attention-level modifications to achieve even finer-grained control over specific aspects of generation.

## B  RELATED WORKS

### B.1  LLM ALIGNMENT

**Alignment Paradigm**   Alignment approaches for LLMs fall into two primary paradigms. Fine-tuning via RLHF [37, 53, 40]—where a reward model guides policy optimization—yields robust performance but depends on a multi-stage loop, which can be resource-intensive [37, 40]. Direct Preference Optimization (DPO) [35] recasts this as a supervised loss, yet still demands significant memory. Inference-time interventions sidestep model updates: prompt engineering can nudge outputs with almost no extra compute [1]. Guided decoding as another effective branch has also been well-explored: ARGS weave reward-model scores into token probabilities [16]. Mudgal et al.[29] and Han et al.[14] train a prefix-based reward scorer to guide generation from a partial hypothesis. DeAL [15], by contrast, casts decoding as an A* search. TransferQ* [6] introduces a quantile-based policy adjustment. Energy-based methods like COLD [34] and BOLT [26] perform iterative gradient-based search at the logit level to find low-energy (high-reward) sequences. However, these methods fundamentally lack a principled mechanism for *precise attribute intensity control*. They are designed to monotonically shift outputs toward preferred extremes—for example, by maximizing a reward function (i.e. ARGS) or minimizing a static energy function (i.e. COLD and BOLT). They are not formulated as a target-reaching problem and thus offer no built-in stopping criterion to hit a specific scalar target (e.g., "helpfulness = 3"), which would require extensive, per-sample hyperparameter tuning.

**Multi-objective alignment**   Another critical direction in LLM alignment is multi-objective alignment, which is crucial for real-world deployment where LLMs must balance competing attributes based on user preferences. Recent works on multi-objective alignment have explored various ways to approximate Pareto-optimal trade-offs. [36] trains separate policies for each reward preference via RLHF and interpolates them post hoc. MODPO [52] extends Direct Preference Optimization to handle multiple objectives. RiC [48] reduces training costs by applying reward-conditioned supervised fine-tuning and lightweight online data augmentation. Panacea [51] further embeds preference vectors into model parameters through SVD-LoRA, enabling a single model to generalize across objectives after training. CPO [13] introduces controllable preference tokens and extends SFT/DPO to condition explicitly on multi-objective preference scores, enabling controllable trade-offs among values such as helpfulness, honesty, and harmlessness. Preference Orchestrator [22] learns a prompt-aware adapter that infers context-specific preference weights to combine multiple reward models, automatically adapting objective trade-offs to each input. PARM [20] tackles multi-objective test-time alignment by training a single preference-aware autoregressive reward model conditioned on preference vectors to guide frozen LLMs along preference trade-offs during decoding. Despite these advancements, these methods still require training new policies or reward models and primarily operate in token-probability space. In contrast, our method achieves efficient multi-objective alignment entirely at test time via value-function–guided representation editing in hidden space, without updating the base LLM parameters.

### B.2  REPRESENTATION ENGINEERING

Activation perturbation began with plug-and-play methods like PPLM [7], which use attribute-specific classifiers to nudge hidden states. However, this approach is often computationally inefficient for large LMs,

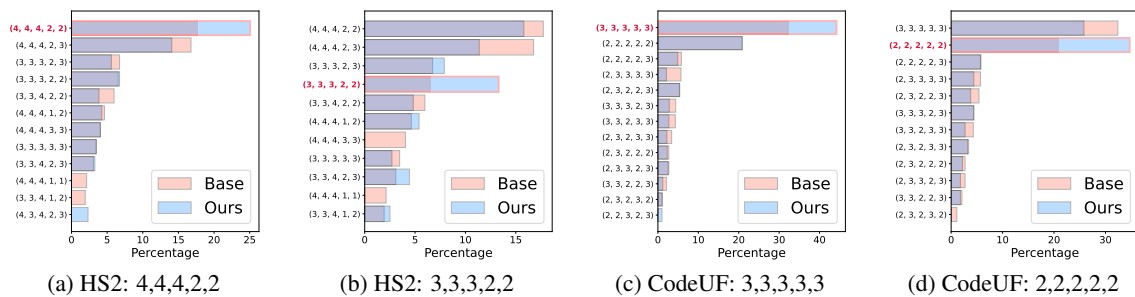

(a) HS2: 4,4,4,2,2     (b) HS2: 3,3,3,2,2     (c) CodeUF: 3,3,3,3,3     (d) CodeUF: 2,2,2,2,2

Figure 5: LLaMA-3.2-3b attribute intensity distributions. Base is the before-intervene distribution, ours is the after-intervene distribution.

as its intervention loop requires repeated forward and backward passes through the expensive LM head to compute gradients from the logit-space loss [5, 27] Subsequent studies showed that both learned and handcrafted steering vectors can control style [38, 41] and that targeting attention-head outputs boosts factual accuracy [19]. [33] applies steering vectors constructed from activation differences. [3] optimizes steering vectors using contrastive preference pairs. Liu et al. further interpret in-context learning as shifting latent states [24]. More recently, representation fine-tuning leverages low-rank projection matrices to edit activations efficiently [46, 45], whereas Liu et al.'s two-phase approach first identifies steering directions via fine-tuning before applying them [25]. Similarly, these methods primarily focus on binary or categorical attribute control. They are not designed for precisely targeting specific attribute intensities on a continuous scale, as they lack the explicit target-reaching objective that provides a principled, per-sample steering signal and stopping criterion.

## C  ADDITIONAL EXPERIMENT RESULTS

### C.1  INTERVENED ATTRIBUTE INTENSITY DISTRIBUTION

Figures 5a and 5b illustrate the attribute-intensity score distributions for both base and PRE-CONTROL under two different intervention targets. PRE-CONTROL not only amplifies the proportion of samples at the originally dominant intensity (4,4,4,2,2) but also effectively shifts the distribution to make the new target (3,3,3,2,2) the prevailing attribute intensity.

### C.2  ITERATIVE INTERVENTION

To enable continuous steering of generation towards user-specified attribute intensity, we feed the model's generation from the previous iteration back and ask to re-address it. Incorporating previous generations as additional context enables more precise steering of the model toward the target output. We reveal our iterative prompt template in Figure 6.

### C.3  FULL INTERVENTION RESULTS

To demonstrate the robustness of our approach, we assess its performance across a range of target attribute intensity scores. The complete results for these varied targets are shown in Table 14 and 15. PRE-CONTROL demonstrates a consistent, strong performance compared to all baselines in various settings.

> **Iterative Intervention Prompt Template**
>
> **[USER INPUT]**
> *User question prompt*
>
> **[ASSISTANT INPUT]**
> *Model response from previous iteration*
>
> **[USER INPUT]**
> Based on the conversation above, please re-address the following question. Begin immediately with the answer content.
> *User question prompt*

Figure 6: Iterative prompting template for attribute intensity control. This is an example of a single-turn conversation. For multi-turn conversation, we could simply add all previous conversations before the user final question prompt.

### C.4 ITERATIVE PARETO FRONTIER APPROXIMATION

In Section 3.3, we show that using PRE-CONTROL to approximate the Pareto frontier yields a stronger frontier. To refine this further, we apply the same interpolation function from the first pass to generate new target points and then reapply PRE-CONTROL. Figures 7a and 7b display the more dominant frontiers obtained after two iterations with linear and upper-convex-hull interpolation, respectively. Figures 8 quantify these iterative frontiers using the metrics defined in Section 3.3. Together, these results demonstrate that iterative approximation with PRE-CONTROL steadily guides the LLM toward increasingly dominant regions of the Pareto surface.

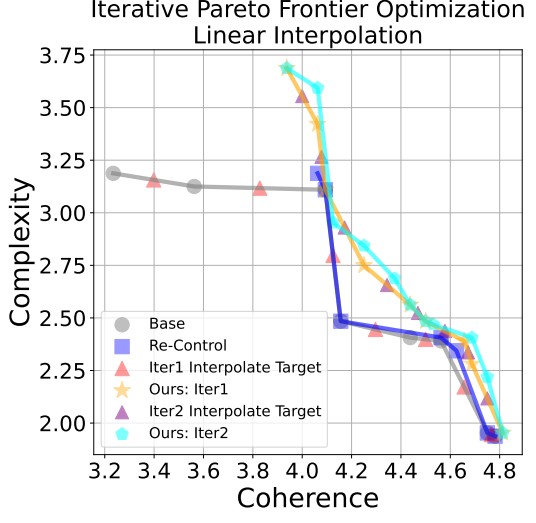
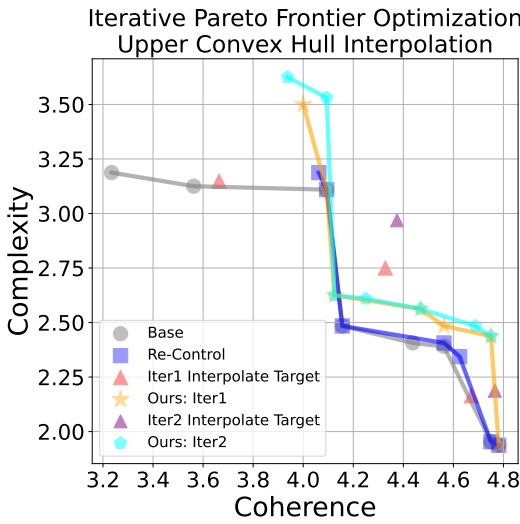

(a) Using linear interpolation targets

(b) Using upper convex-hull targets

Figure 7: Iterative Pareto frontier approximation after two iterations.

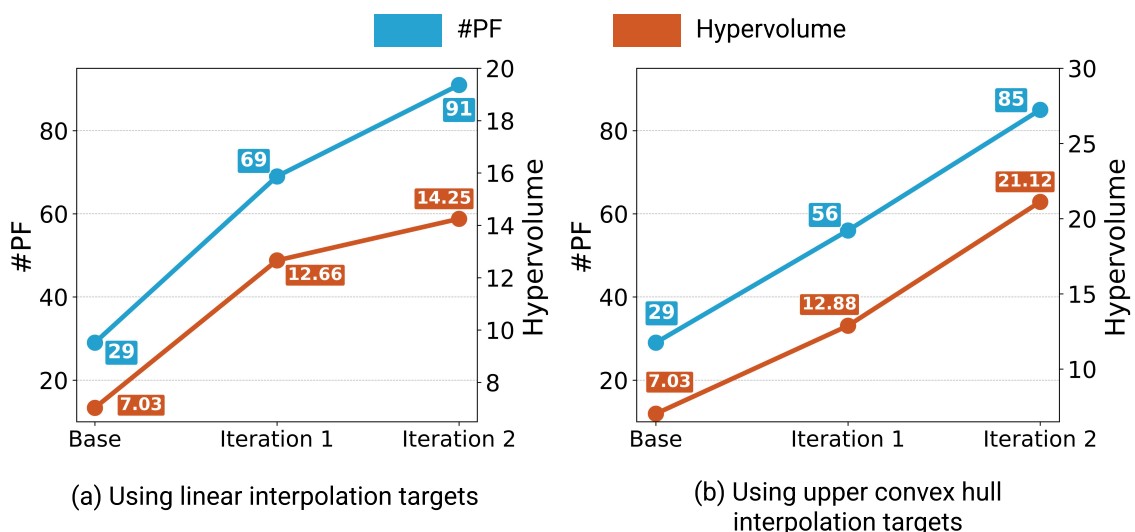

Figure 8: Quantitative results of iterative Pareto frontier approximation after two iterations.

| Method | HV | # Samples | Overhead (GPUh) |
|---|---|---|---|
| Base | 7.03 | – | – |
| Best-of-$N$ | 15.27 | 50k | 7.8 |
| **PRE-CONTROL** | **16.81** | 15k | **2.1** |

Table 3: Controllable distillation quality and efficiency.

## C.5 CONTROLLABLE DISTILLATION

Table 3 presents results from our controllable distillation experiment, where we aim to develop intervention-free aligned models by training on high-quality preference-controlled samples. Our results demonstrate PRE-CONTROL achieves better performance with significantly lower resources. With only 15k samples and 2.1 GPU hours of computation, PRE-CONTROL attains a higher hypervolume (16.81) than Best-of-N (BoN) distillation (15.27), which requires 50k samples and 7.8 GPU hours. This represents a 3.3× reduction in sample requirement and 3.7× decrease in computational overhead while still improving quality. The efficiency advantage stems from PRE-CONTROL's ability on directly generating high-quality training examples at specific attribute intensity, as opposed to BoN's approach of generating much candidate samples and filtering, which incurs substantial costs.

## D EXPERIMENTAL DETAILS

### D.1 HELPSTEER2

We evaluate our method on HelpSteer2 dataset, which is a widely used multi-attribute preference dataset for LLM alignment. This dataset comprises 20,324 training samples and 1,038 test samples. Each prompt is paired with two annotated responses, evaluated across five attributes: *helpfulness*, *correctness*, *coherence*,

*complexity*, and *verbosity* by a scale from 0 to 4. We adopt LLaMA-3.2-3b [12] and Phi-4-mini [28] as our base instructed fine-tuned AI assistants to generate text responses based on the prompts from HelpSteer2. Therefore, our training and test data sizes are 10162 and 519. Adhering to the standard practice, we set the maximum lengths of the prompt and maximum output token length to 2048 and 512, respectively.

## D.2 CODE-ULTRAFEEDBACK

To further evaluate our method, we adopt Code-UltraFeedback, a multi-attribute code preference dataset. The dataset consists of 10,000 complex instructions. Each is paired with four LLMs responses aligned with five coding preferences: *code explanation*, *code complexity and efficiency*, *code readability*, *coding style*, and *instruction-following*. Similar to HelpSteer2 experiment, we use LLaMA-3.2-3b and Phi-4-mini as our base instructed fine-tuned AI assistants. We randomly sample 1,000 instructions from Code-UltraFeedback to be our test set for evaluating these models. Therefore, our training and test data sizes are 9,000 and 1,000. We set the maximum lengths of the prompt and maximum output token length to 2048 and 1024, respectively.

## D.3 IMPLEMENTATION DETAILS

**Validation Set.** We randomly sample 10% of the training data to construct a separate validation set for selecting the hyperparameter — the step size $\alpha$ — based on success rate.

**Reward model.** For the reward model, we use ArmoRM-Llama3-8B [42], which is trained on several multi-attribute alignment datasets, including both HelpSteer2 and Code-UltraFeedback. We use a batch size of 256 to evaluate LLM-generated responses.

**Attribute weight $w$.** For attribute weight in Equation 9, we set $w_i = 1, \forall i$ empirically.

**PRE-CONTROL.** To construct the training dataset for the value function, we apply greedy decoding to sample one response per prompt from HelpSteer2 and Code-UltraFeedback ($M = 1$). The value function is trained on the last layer of the hidden states $h_t$. At test time, we inject multi-attribute control signals solely to this layer as well. We parameterize the value function as a three-layer neural network for both LLaMA-3.2-3b and Phi-4-mini. We use Adam [17] as our value function training optimizer. We adopt early stopping techniques to train the value function. Training stops when the test loss fails to improve for a specified number of consecutive epochs (the `patience` hyperparameter in Table 4). Table 4 presents the training hyperparameters. Figure 9 depicts the training loss of our value function, demonstrating its convergence. Table 5 presents the inference hyperparameters. Because our intervention is closed-form and driven by a target attribute intensity, it doesn't rely on a fixed number of updates. Instead, we halt once the value-function output on the hidden states falls within a specified tolerance of that target for a specified number of consecutive epochs.

**Static Representation** Following [19][2], we train a linear regression layer on top of LLM's hidden state to predict the expected reward. At inference, we shift activations along the weight direction using intervention strength $\alpha$, selected via validation set optimization.

**Multi-Attribute Steer** We use the codebase[3] from [31]. For each attribute in both HelpSteer2 and Code-UltraFeedback, we randomly select 1000 positive samples and 1000 negative samples to learn the steering vectors. We adopt the same design by classifying samples with scores of 3 or 4 as positive and samples with scores < 3 as negative (on a 0-4 scale).

---

[2] https://github.com/likenneth/honest_llama
[3] https://github.com/duykhuongnguyen/MAT-Steer

**Prompting engineering.** Following [30], we instruct the model to provide responses that align with the specified attribute intensity. For HelpSteer2, we use the attribute definition as listed in its Hugging-Face repository. For Code-UltraFeedback, we adopt the attribute definition in [44]. Figure 10 and 11 show our prompt template.

**Representation Editing** We use the codebase[a] from [18]. We set the value function architecture exactly the same as ours, and train it using REControl's objective. We limit the number of updates to 100. Training and inference hyperparameters for REControl are summarized in Table 6 and Table 7, respectively.

[a]`https://github.com/Lingkai-Kong/RE-Control`

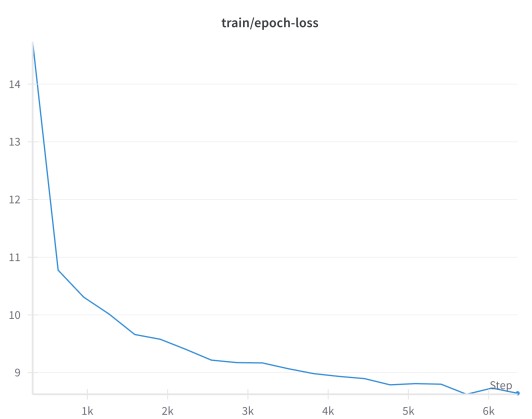

Figure 9: Value function training loss curve

| Backbone | Hyperparameter | HelpSteer2 Value | Code-UltraFeedback Value |
|---|---|---|---|
| **LLaMA-3.2-3b** | Number of epochs | 100 | 100 |
| | Learning rate | $1 \times 10^{-4}$ | $1 \times 10^{-4}$ |
| | Batch size | 32 | 32 |
| | Floating point format | `fp16` (Half-precision) | `fp16` (Half-precision) |
| | Number of layers | 3 | 3 |
| | Hidden dimension | 3072 | 3072 |
| | $\lambda$ | 0.9 | 0.9 |
| | Number of Patience | 10 | 10 |
| **Phi-4-mini** | Number of epochs | 100 | 100 |
| | Learning rate | $1 \times 10^{-4}$ | $1 \times 10^{-4}$ |
| | Batch size | 64 | 32 |
| | Floating point format | `fp16` (Half-precision) | `fp16` (Half-precision) |
| | Number of layers | 3 | 3 |
| | Hidden dimension | 3072 | 3072 |
| | $\lambda$ | 0.9 | 0.9 |
| | Number of Patience | 10 | 10 |

Table 4: Summary of hyperparameters used in training the value function of PRE-CONTROL.

**Controllable Distillation** Table D.3 summarizes our hyperparameters for our controllable distillation experiments. We apply the same hyperparameters for both Best-of-N distillation and our Pareto frontier distillation.

| Backbone | Hyperparameter | HelpSteer2 Value | Code-UltraFeedback Value |
|---|---|---|---|
| LLaMA-3.2-3b | Step size | $7 \times 10^{-2}$ | $9 \times 10^{-3}$ |
| | Batch size | 24 | 12 |
| | Floating point format | `fp16` (Half-precision) | `fp16` (Half-precision) |
| | Max generation length | 512 | 1024 |
| | Weight Decay | 0.01 | 0.01 |
| | Minimum $\Delta$ | $1 \times 10^{-4}$ | $1 \times 10^{-4}$ |
| | Number of Patience | 10 | 10 |
| | Tolerance | $1 \times 10^{-4}$ | $1 \times 10^{-4}$ |
| Phi-4-mini | Step size | $8 \times 10^{-4}$ | $3 \times 10^{-3}$ |
| | Batch size | 24 | 12 |
| | Floating point format | `fp16` (Half-precision) | `fp16` (Half-precision) |
| | Max generation length | 512 | 1024 |
| | Weight Decay | 0.01 | 0.01 |
| | Minimum $\Delta$ | $1 \times 10^{-4}$ | $1 \times 10^{-4}$ |
| | Number of Patience | 10 | 10 |
| | Tolerance | $1 \times 10^{-4}$ | $1 \times 10^{-4}$ |

Table 5: Summary of hyperparameters of PRE-CONTROL at test time.

---

**HelpSteer2 Attribute Intensity Control Prompt Template**

**[SYSTEM INPUT]**
You are an AI assistant tasked with generating a high-quality response that will be evaluated across multiple attributes. Each attribute is scored from 0 to 4 according to the following general scale:

- 0: Completely unsatisfactory — does not demonstrate any relevant quality.
- 1: Poor — minimal expression of the intended quality; largely ineffective.
- 2: Fair — partially demonstrates the desired quality, but with notable limitations.
- 3: Good — mostly meets expectations; minor gaps or inconsistencies.
- 4: Excellent — fully meets the intended standard; consistent, complete, and high-quality.

The evaluation attributes for this task are:
Helpfulness: Overall helpfulness of the response to the prompt.
Correctness: Inclusion of all pertinent facts without errors.
Coherence: Consistency and clarity of expression.
Complexity: Intellectual depth required to write the response (i.e., whether the response could be written by anyone with basic language competency or requires deep domain expertise).
Verbosity: Amount of detail included in the response, relative to what is asked for in the prompt.

Your goal is to write a response that would receive a score of {target_attribute_score} in Helpfulness, Correctness, Coherence, Complexity, and Verbosity, respectively

**[USER INPUT]**
......

Figure 10: HelpSteer2 prompting template for attribute intensity control

| Backbone | Hyperparameter | Value |
|---|---|---|
| LLaMA-3.2-3B | Step size | $2 \times 10^{-5}$ |
| | Number of updates | 3 |
| | Batch size | 128 |
| | Floating point format | `fp16` (Half-precision) |

Table 8: Summary of hyperparameters of controllable distillation.

---

**Code-UltraFeedback Attribute Intensity Control Prompt Template**

**[SYSTEM INPUT]**
You are an AI assistant tasked with generating a high-quality response that will be evaluated across multiple attributes. Each attribute is scored from 0 to 4 according to the following general scale:

- 0: Completely unsatisfactory — does not demonstrate any relevant quality.
- 1: Poor — minimal expression of the intended quality; largely ineffective.
- 2: Fair — partially demonstrates the desired quality, but with notable limitations.
- 3: Good — mostly meets expectations; minor gaps or inconsistencies.
- 4: Excellent — fully meets the intended standard; consistent, complete, and high-quality.

The evaluation attributes for this task are:
Code Complexity and Efficiency: Generating code optimized for performance in terms of speed and resource utilization.
Style: Writing code that not only meets syntactical correctness but also aligns with the idiomatic practices and stylistic norms of the programming language.
Code Explanation: Generating code with detailed natural language explanations. It underscores the importance of an LLM in providing a solution with explanations that can serve as a bridge between potentially complex code and users while improving trustworthiness.
Instruction-following: Strict adherence of the LLM to the instructions provided by users. This attribute is foundational for ensuring that LLMs truly follow the user intent and thus provide personalized responses to instructions.
Code Readability: Clarity and understandability of the code itself through its structure, style, and the presence of meaningful documentation and in-line comments.

Your goal is to write a response that would receive a score of `{target_attribute_score}` in Code Complexity and Efficiency, Style, Code Explanation, Instruction-following, and Code Readability, respectively

**[USER INPUT]**
......

Figure 11: Code-UltraFeedback prompting template for attribute intensity control

| Backbone | Hyperparameter | HelpSteer2 Value | Code-UltraFeedback Value |
|----------|----------------|------------------|--------------------------|
| **LLaMA-3.2-3b** | Number of epochs | 100 | 100 |
| | Learning rate | $1 \times 10^{-4}$ | $1 \times 10^{-4}$ |
| | Batch size | 32 | 32 |
| | Floating point format | `fp16` (Half-precision) | `fp16` (Half-precision) |
| | Number of layers | 3 | 3 |
| | Hidden dimension | 3072 | 3072 |
| **Phi-4-mini** | Number of epochs | 100 | 100 |
| | Learning rate | $1 \times 10^{-4}$ | $1 \times 10^{-4}$ |
| | Batch size | 32 | 32 |
| | Floating point format | `fp16` (Half-precision) | `fp16` (Half-precision) |
| | Number of layers | 3 | 3 |
| | Hidden dimension | 3072 | 3072 |

Table 6: Summary of hyperparameters used in training the value function of REControl.

## D.4    PARETO FRONTIER INTERPOLATION FUNCTION

We introduce an $\alpha$-weighted interpolation scheme to enrich the Pareto frontier with synthetic target points, thereby improving frontier coverage. Throughout, let $\mathcal{P} = \{ \mathbf{p}_i \in \mathbb{R}^k \}_{i=1}^N (k \leq 5$ in our experiments) be the

| Backbone | Hyperparameter | HelpSteer2 Value | Code-UltraFeedback Value |
|---|---|---|---|
| **LLaMA-3.2-3b** | Step size | $1 \times 10^{-3}$ | $5 \times 10^{-4}$ |
| | Number of updates | 100 | 100 |
| | Batch size | 24 | 12 |
| | Floating point format | `fp16` (Half-precision) | `fp16` (Half-precision) |
| | Max generation length | 512 | 1024 |
| | Weight Decay | 0.01 | 0.01 |
| **Phi-4-mini** | Step size | $1 \times 10^{-3}$ | $1 \times 10^{-3}$ |
| | Number of updates | 100 | 100 |
| | Batch size | 24 | 12 |
| | Floating point format | `fp16` (Half-precision) | `fp16` (Half-precision) |
| | Max generation length | 512 | 1024 |
| | Weight Decay | 0.01 | 0.01 |

Table 7: Summary of hyperparameters of REControl at test time.

ordered frontier. Denote the coordinates of any point by $\mathbf{p} = (x_1, \dots, x_j)^\top, 2 \le j \le 5, j \in \mathbb{Z}$. Below, we detail the two interpolation functions we employ.

### D.4.1 LINEAR INTERPOLATION

Our linear interpolation function is a local $\alpha$-neighbor interpolator that densifies the frontier between consecutive points. For each pair of consecutive samples $\mathbf{p}_i$, $\mathbf{p}_{i+1}$ we add an $\alpha$-weighted interior point

$$\mathbf{m}_i^{(\alpha)} = \alpha\,\mathbf{p}_i + (1-\alpha)\,\mathbf{p}_{i+1}, \qquad i = 1, \dots, N-1,\ \alpha \in (0,1). \tag{13}$$

Our synthetic targets would then be $\{\mathbf{m}_i^{(\alpha)}\}_{i=1}^{N-1}$. Empirically, we set $\alpha = 0.5$.

### D.4.2 UPPER CONVEX HULL INTERPOLATION

Another interpolation function we implement is an $\alpha$-upper-hull interpolator that preserves global concavity and Pareto dominance. To maintain a *globally concave* frontier, we first extract the upper convex hull

$$\mathcal{H} = \{\mathbf{v}_j\}_{j=1}^M = \mathrm{vert}\big(\mathrm{conv}\{\mathbf{p}_i\}_{i=1}^N\big),$$

where

$$\mathrm{conv}\{\mathbf{p}_i\}_{i=1}^N = \Big\{\sum_{i=1}^N \lambda_i\,\mathbf{p}_i \ \Big|\ \lambda_i \ge 0,\ \sum_{i=1}^N \lambda_i = 1\Big\}.$$

We then interpolate between consecutive hull vertices:

$$\mathbf{m}_j^{\mathcal{H},(\alpha)} = \alpha\,\mathbf{v}_j + (1-\alpha)\,\mathbf{v}_{j+1}, \qquad j = 1, \dots, M-1. \tag{14}$$

Because $\mathbf{m}_j^{\mathcal{H},(\alpha)}$ lies on the segment $[\mathbf{v}_j, \mathbf{v}_{j+1}]$, the augmented set $\mathcal{H} \cup \{\mathbf{m}_j^{\mathcal{H},(\alpha)}\}$ remains concave and dominates all interior points:

$$y\Big(\mathbf{m}_j^{\mathcal{H},(\alpha)}\Big) \ \ge \ y(\mathbf{p}_i), \quad \forall\,\mathbf{p}_i \in \mathcal{P}.$$

Our synthetic targets would then be $\{\mathbf{m}_j^{\mathcal{H},(\alpha)}\}_{j=1}^{M-1}$. Empirically, we set $\alpha = 0.5$.

# E    COMPUTING INFRASTRUCTURE

We conduct our experiments on a server equipped with 4 NVIDIA A100 (80GB VRAM) GPUs. We utilize the NVIDIA CUDA toolkit version 12.4. All experiments are implemented using Python 3.10.4, the PyTorch framework version 2.3.1, and the Transformer library version 4.51.3.

# F    LATENCY

We analyze the computational efficiency of PRE-CONTROL by evaluating its two primary costs: the one-time, offline training of the value function and online inference-time intervention.

**Value Function Training**    Training a 4-layer neural network on 1,0162 samples requires only 0.34 GPU hours. This modest, one-time cost makes the value function practical to train and update.

**Inference-Time Intervention**    To quantify the computational overhead of test-time intervention, we compared the cost of generating 434 test samples from HelpSteer2 using LLaMA-3.2-3b against all baselines. We present the results in Table 9. While PRE-CONTROL incurs moderate overhead due to token-level optimization, it remains within the same order of magnitude as other baselines. Given its strong performance in multi-attribute controllability, this overhead represents a practical trade-off.

Since generation steps (output token length) and the degree of parallelization (batch size) significantly impact practical computational cost, we analyzed how overhead scales with each factor independently. For output token length, we keep batch size the same to 24. For batch size, we keep the output token length the same to 512. Results in Table 10 indicates that the inference cost scales linearly with the output token length while also benefiting significantly from parallelization, as the total cost decreases with a larger batch size.

| Method | GPU Hours |
|--------|-----------|
| Base | 0.02 |
| Prompting | 0.02 |
| ITI | 0.06 |
| MAT-Steer | 0.07 |
| RE-Control[4] | 0.10 |
| PRE-CONTROL | 0.09 |

Table 9: Computational Cost Comparison of LLaMA-3.2-3b on HelpSteer2

| Output Token Length | GPU Hours |
|---------------------|-----------|
| 256 | 0.04 |
| 512 | 0.09 |
| 768 | 0.14 |

| Batch Size | GPU Hours |
|------------|-----------|
| 12 | 0.14 |
| 24 | 0.09 |
| 48 | 0.05 |

Table 10: Computational Cost on Different Output Token Length and Batch Size

# G    CASE STUDY

In Table 11 and Table 13, we present qualitative examples demonstrating PRE-CONTROL's ability to precisely control attribute intensities.

**Negative Target Scenario.** The base model produces an overly detailed response scoring [4,4,4,3,3], featuring extensive component-by-component breakdowns followed by comprehensive summaries. While thorough,

---

[4]RE-Control's computational cost is sensitive to the number of intervention steps. We follow [18] to use 100 steps which yields the best performance in terms of reward scores.

such verbosity may overwhelm users seeking quick answers. We therefore set target scores of [3,3,3,2,2], intending to reduce both complexity and verbosity while slightly moderating other attributes for a more concise response. PRE-CONTROL successfully steers the generation to match these targets, producing a deliberately streamlined output that removes granular details, eliminates redundant summaries, and presents only essential information—demonstrating precise control even when reducing attribute intensities.

**Positive Target Scenario 1.** The base model generates a response with scores [4,4,4,1,1] for *helpfulness*, *coherence*, *correctness*, *complexity*, and *verbosity*, respectively. While the response is helpful and correct, it lacks detail—providing only minimal explanations without elaborating on command options or their purposes. To address this deficiency, we set target scores of [4,4,4,2,2], aiming to maintain the high quality while increasing both complexity and verbosity to provide more comprehensive information. After applying PRE-CONTROL, the model successfully achieves these exact target scores by enriching the response with explicit clarifications of command flags, detailed option descriptions, and expanded explanations of each component's purpose.

**Positive Target Scenario 2.** The base model produces a functionally correct solution with scores [3,3,2,3,3] for *complexity and efficiency*, *style*, *explanation*, *instruction-following*, and *readability*, but its explanation of the code is brief and offers limited guidance beyond the raw implementation. We set a higher target on the explanation dimension and apply PRE-CONTROL, which yields a response with scores [3,3,3,3,3] by enriching the answer with an explicit step-by-step explanation clarifying the function's behavior and usage. This example illustrates that PRE-CONTROL can selectively improve explanatory quality while preserving the underlying correctness and overall structure of the original solution.

## H HYPERPARAMETER STUDY

To better understand the characteristics of PRE-CONTROL, we analyze the sensitivity of its key hyperparameter, intervention step size $\alpha$ (in Eq. 9), which controls the magnitude of the gradient update at each intervention step. We vary $\alpha$ across several orders of magnitude and measure the impact on the overall Success Rate. We present the result in Figure 12. Success Rate remains stable, fluctuating between 6.51% and 6.99% for $\alpha$ values ranging from $10^{-1}$ down to $10^{-3}$. This demonstrates that PRE-CONTROL is robust across a wide range of step size values.

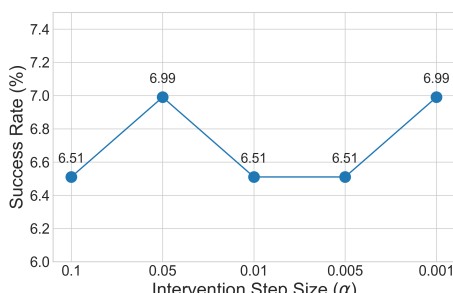

Figure 12: Sensitivity of intervention step size $\alpha$.

---

**Negative Target Example**

**Prompt:**

USER: What is the meaning of "Redundant UPS on all critical infrastructure in PiXELL-River datacentre with capacity to expand by approximately 30%"?

---

**Base**

A very technical question.
Here's a breakdown of what it means:

- **Redundant UPS**: UPS stands for Uninterruptible Power Supply. It's a backup power system that provides power to critical infrastructure in the event of a power failure. A redundant UPS means that there are multiple UPS systems in place, which can take over in case one of them fails or is taken offline for maintenance.
- **All critical infrastructure**: This refers to the essential systems and equipment that support the operation of the datacentre, such as:
  - Power distribution units (PDUs)
  - Network equipment (e.g., routers, switches)
  - Server racks and cabinets
  - Cooling systems
  - Security systems
- **PiXELL-River datacentre**: This is the specific datacentre where the redundant UPS system is located.
- **Capacity to expand by approximately 30%**: This means that the redundant UPS system is designed to support an additional 30% of the datacentre's power requirements, allowing for future growth and expansion of the facility.

In summary, the statement means that the PiXELL-River datacentre has a redundant UPS system in place to ensure continuous power supply to all critical infrastructure, with a capacity to expand by approximately 30% to accommodate future growth.

This is a critical aspect of datacentre design, as it ensures that the facility can continue to operate reliably and efficiently, even in the event of a power failure or other disruptions.

---

**PRE-CONTROL**

The PiXELL-River datacentre has implemented redundant uninterruptible power supply (UPS) systems across its critical infrastructure to ensure continuous operation in case of power failures. Additionally, the datacentre has a capacity to expand its power capacity by approximately 30% by adding more UPS systems, allowing it to accommodate future growth without significant upgrades or replacements.

---

Table 11: Qualitative examples of negative target score showing the alignment performance of PRE-CONTROL. **Base** response has a score of 4,4,4,3,3. **PRE-CONTROL** response has a score of 3,3,3,2,2.

**Positive Target Example 1**

**Prompt:**

USER: how to run airflow locally
ASSISTANT: Here are the steps for running Airflow locally: 1. Install Python and pip. 2. Install Airflow by running: `pip install airflow` 3. Create a directory for your Airflow installation, and navigate to it in your terminal. 4. Run: `airflow initdb` to create the Airflow database. 5. `airflow webserver` to start the Airflow web server.

USER: is this for mac right?
ASSISTANT: Yes, these are the steps for running Airflow locally on a Mac. You will need to install Python and pip, and then you can install Airflow using pip install airflow. Next, create a directory for your Airflow installation and navigate to it in your terminal, then run airflow initdb to create the Airflow database. Finally, run airflow webserver to start the Airflow web server.

USER: can you describe it without pip?
ASSISTANT: Here are the steps for running Airflow locally on a Mac without using `pip`: 1. Install Python. 2. Create a directory for your Airflow installation. 3. Navigate to the directory you created in step 2 in your terminal. 4. Run `airflow initdb` to create the Airflow database. 5. Run airflow webserver to start the Airflow web server.

USER: what's the username and password the webserver asks
ASSISTANT: The username and password the webserver asks for is the username and password for your Airflow database. You can create a new user for your Airflow database using the `airflow create_user` command. The username and password for the new user will be prompted for when you run this command.

USER: can you show me an example command on creating a user
ASSISTANT:

**Base**

Here's an example command for creating a new user in Airflow:

```
airflow users create --username myuser \
                     --password mypassword \
                     --firstname My \
                     --lastname User
```

This command creates a new user with the username `myuser`, password `mypassword`, and the first and last names "My User."

**PRE-CONTROL**

Here is an example of a command to create a new user in Airflow:

```
airflow db init
airflow create_user -u myuser -p mypassword -r myrole
```

Here's what each part of the command does:
- `airflow db init` initializes the Airflow database.
- `airflow create_user` creates a new user.
- `-u myuser` sets the username for the new user to "myuser."
- `-p mypassword` sets the password for the new user to "mypassword."
- `-r myrole` sets the role for the new user to "myrole."

Note: You can also use `airflow create_user` with the `-username` and `-password` options instead of `-u` and `-p`:

```
airflow create_user --username myuser \
                    --password mypassword \
                    --role myrole
```

Table 12: Qualitative examples of positive target score showing the alignment performance of PRE-CONTROL. **Base** response has a score of 4,4,4,1,1. **PRE-CONTROL** response has a score of 4,4,4,2,2.

**Positive Target Example 2**

**Prompt:**

USER: Author a Python procedure labeled ćoncatenateAlphabetsẃhich accepts two parameters of the data type string and outputs the resultant string, amalgamated from both inputs. string1 = "Hello" string2 = "World"
ASSISTANT:

**Base**

To create a Python procedure named 'concatenateAlphabets' that takes two string parameters and returns their concatenated result, you can define a function as follows:

```python
def concatenateAlphabets(string1, string2):
    """
    Concatenates two input strings and returns the result.

    Parameters:
    string1 (str): The first input string.
    string2 (str): The second input string.

    Returns:
    str: The concatenated result of string1 and string2.
    """
    return string1 + string2
# Example usage:
string1 = "Hello"
string2 = "World"
result = concatenateAlphabets(string1, string2)
print(result)  # Output: HelloWorld
```

This function simply uses the '+' operator to concatenate the two input strings and returns the combined result. You can test it with the provided example strings "Hello" and "World", which will output "HelloWorld".

**PRE-CONTROL**

To create a Python function named 'concatenateAlphabets' that takes two string parameters and returns their concatenated result, you can follow the code example below. This function will simply use the '+' operator to concatenate the two input strings.
Here's how you can implement it:

```python
def concatenateAlphabets(string1, string2):
    """
    Concatenates two input strings and returns the result.

    Parameters:
    string1 (str): The first input string.
    string2 (str): The second input string.

    Returns:
    str: The concatenated result of string1 and string2.
    """
    return string1 + string2

# Example usage:
string1 = "Hello"
string2 = "World"
result = concatenateAlphabets(string1, string2)
print(result)  # Output: HelloWorld
```

Explanation:
- The function 'concatenateAlphabets' is defined with two parameters, 'string1' and 'string2'.
- Inside the function, the '+' operator is used to concatenate 'string1' and 'string2'.
- The concatenated result is returned.
- An example usage is provided to demonstrate how the function works with the strings '"Hello"' and '"World"', producing the output '"HelloWorld"'.

Table 13: Qualitative examples of positive target score showing the alignment performance of PRE-CONTROL. **Base** response has a score of 3,3,2,3,3. **PRE-CONTROL** response has a score of 3,3,3,3,3.

| Backbone | | | Llama-3.2-3B | | |
|---|---|---|---|---|---|
| **Dataset** | **Target Score** | **Method** | **Diversity ↓** | **$\ell_1$ Distance to Target ↓** | **Success Rate (%) ↑** |
| HelpSteer2 | 4,4,4,2,2 | Base | 0.626 | 2.19 | N/A |
| | | Prompting | 0.941 | 2.17 | 5.39 |
| | | ITI | 0.604 | 3.02 | 3.75 |
| | | Re-Control | 0.946 | **2.16** | 5.39 |
| | | MAT-Steer | 0.739 | 2.22 | 5.17 |
| | | **Ours** | **0.558** | **2.16** | **7.96** |
| | 4,4,4,3,3 | Base | 0.695 | 3.12 | N/A |
| | | Prompting | 0.930 | 3.12 | 1.20 |
| | | ITI | 0.513 | 3.09 | 0.80 |
| | | Re-Control | 0.931 | 3.07 | 1.00 |
| | | MAT-Steer | 0.487 | 3.05 | 1.36 |
| | | **Ours** | **0.440** | **3.02** | **1.81** |
| | 3,3,3,2,2 | Base | 0.656 | 2.76 | N/A |
| | | Prompting | 0.987 | 2.73 | 2.47 |
| | | ITI | **0.294** | 2.69 | 5.48 |
| | | Re-Control | 0.986 | 2.72 | 2.27 |
| | | MAT-Steer | 0.539 | **2.57** | 5.84 |
| | | **Ours** | 0.251 | 2.63 | **6.60** |
| Code-UltraFeedback | 3,3,3,3,3 | Base | 0.876 | 2.29 | N/A |
| | | Prompting | 0.879 | 2.21 | 6.80 |
| | | ITI | 0.741 | 2.62 | 12.72 |
| | | Re-Control | 0.880 | 2.21 | 7.54 |
| | | MAT-Steer | 0.778 | 2.41 | 13.63 |
| | | **Ours** | **0.614** | **2.08** | **17.46** |
| | 2,3,3,2,3 | Base | 0.838 | 2.24 | N/A |
| | | Prompting | 0.838 | 2.23 | 1.85 |
| | | ITI | 0.670 | 2.33 | 1.82 |
| | | Re-Control | 0.831 | 2.18 | 2.06 |
| | | MAT-Steer | 0.587 | 2.38 | 1.64 |
| | | **Ours** | **0.512** | 2.17 | **2.77** |
| | 2,2,2,2,2 | Base | 0.874 | 2.95 | N/A |
| | | Prompting | 0.865 | 2.85 | 6.06 |
| | | ITI | 0.441 | 2.83 | 6.79 |
| | | Re-Control | 0.856 | 2.78 | 6.57 |
| | | MAT-Steer | 0.480 | 2.59 | 16.67 |
| | | **Ours** | **0.440** | **1.95** | **30.68** |

Table 14: Comprehensive results for **LLaMA-3.2-3b** with various target scores.

| Backbone | | | Phi-4-mini | | |
|---|---|---|---|---|---|
| **Dataset** | **Target Score** | **Method** | **Diversity** ↓ | **$\ell_1$ Distance to Target** ↓ | **Success Rate (%)** ↑ |
| HelpSteer2 | 4,4,4,2,2 | Base | 0.701 | 2.46 | N/A |
| | | Prompting | 0.698 | 2.42 | 5.23 |
| | | ITI | 0.534 | 3.63 | 2.61 |
| | | Re-Control | 0.611 | 2.51 | 5.70 |
| | | MAT-Steer | 0.503 | 2.46 | 5.48 |
| | | **Ours** | **0.530** | **2.41** | **8.31** |
| | 3,3,3,2,2 | Base | 0.659 | 2.76 | N/A |
| | | Prompting | 0.664 | 2.67 | 5.18 |
| | | ITI | 0.450 | 2.73 | 4.02 |
| | | Re-Control | 0.494 | 2.56 | 5.80 |
| | | MAT-Steer | 0.308 | 2.86 | 8.73 |
| | | **Ours** | **0.291** | **2.46** | **9.11** |
| | 4,3,4,2,3 | Base | 0.632 | 2.78 | N/A |
| | | Prompting | 0.639 | 2.72 | 0.59 |
| | | ITI | 0.565 | 3.50 | 0.59 |
| | | Re-Control | **0.483** | 2.69 | 0.99 |
| | | MAT-Steer | 0.637 | 2.91 | 0.97 |
| | | **Ours** | 0.544 | **2.63** | **2.17** |
| Code-UltraFeedback | 3,3,3,3,3 | Base | 0.902 | 1.57 | N/A |
| | | Prompting | 0.903 | 1.47 | 9.46 |
| | | ITI | 0.789 | 1.55 | 16.49 |
| | | Re-Control | 0.786 | 1.43 | 17.25 |
| | | MAT-Steer | 0.700 | 1.43 | 18.92 |
| | | **Ours** | **0.755** | **1.33** | **24.12** |
| | 2,3,2,2,3 | Base | 0.907 | 2.50 | N/A |
| | | Prompting | 0.906 | 2.51 | 0.72 |
| | | ITI | 0.647 | 2.50 | 1.33 |
| | | Re-Control | 0.570 | 2.48 | 2.46 |
| | | MAT-Steer | 0.586 | 2.49 | 1.89 |
| | | **Ours** | **0.454** | **2.42** | **2.66** |
| | 2,2,2,2,2 | Base | 0.868 | 3.65 | N/A |
| | | Prompting | 0.869 | 3.64 | 2.15 |
| | | ITI | 0.623 | 3.66 | 4.54 |
| | | Re-Control | 0.614 | 3.53 | 6.92 |
| | | MAT-Steer | 0.318 | 2.89 | 8.38 |
| | | **Ours** | **0.322** | **2.58** | **24.83** |

Table 15: Comprehensive results for **Phi-4-mini** with various target scores.

