# OpenReview forum: "Precise Attribute Intensity Control in Large Language Models via Targeted Representation Editing"
_ICLR.cc/2026/Conference — ICLR 2026 Conference Desk Rejected Submission_

### Official Review · Reviewer_SNhh · 2025-10-26

**Soundness:** 2
**Presentation:** 3
**Contribution:** 2
**Rating:** 4
**Confidence:** 4

**Summary:**

This work investigates the use of representation editing for attribute intensity control in large language models using scalar intensities. Authors propose an approach dubbed Pre-Control, which optimizes a MLP-based value function to estimate the (multi-)attribute intensity of resulting outputs from the LLM's hidden representations of partial generations. Once trained, the value function is exploited during generation to perform test-time interventions on LLM representations, aiming to achieve the requested attribute conditioning by adjusting the hidden states via gradient descent. Notably, the traditional alignment problem aiming to maximize/minimize attributes is reformulated as target-reaching to enable finer-grained control over attribute intensities. Authors propose using Pre-Control to efficiently estimate the best combinations of attribute control without incurring in the costly sampling of grid search for various attribute intensity values. Pre-Control is evaluated on LLaMA-3.2-3b and Phi-4-mini for the HelpSteer2 and Code-UltraFeedback datasets, showing improved attribute adherence across one-shot and iterated interventions, and applications to efficient Pareto frontier approximation and controllable model distillation.

**Strengths:**

Originality: The use of TD learning for value function training in the context of LLM attribute control is a creative application of reinforcement learning techniques.

Quality: The experimental evaluation is sufficiently comprehensive, spanning two datasets, two models and various baseline methods. Results seem to suggest consistent improvements over the tested baseline approaches.

Clarity: The mathematical formulations and figures in the paper are clear and well-structure, aiding understanding. The inclusion of qualitative examples in Tables 9-10 helps demonstrate the practical impact of the method.

Significance: The work addresses an important are of research, aiming to render the process of multi-attribute control more efficient. A reduction in sample requirements for distillation while maintaining quality could have significant implications for efficient model adaptation to arbitrary styles, making the proposed implementation desirable over more costly approaches.

**Weaknesses:**

Limited novelty: The proposed method closely resembles to the established Plug-and-play language modeling (PPLM) paradigm [1], which employs the prediction of a conditional attribute model at inference time to update the activations of a language model using a gradient update. The main innovations in this context are 1) the use of temporal difference learning for training the attribute model, as opposed to BoW/supervised classifiers used in PPLM; 2) The usage of scalar values expressing intensity for the property of interest, instead of binary properties to maximize. The combination of various attribute models at inference time was already proposed and evaluated by the PPLM authors, so it cannot be considered particularly novel. For 1), authors state that TD provides "crucial intermediate feedback signals that were previously missing in preference alignment methods". In this context, an ablation of the effectiveness of TD compared to traditional supervised approaches would have been necessary to provide evidence in support of this statement. In the case of 2), requiring attributes to match a specific scalar value can be equated to a progressive shift towards the desired attribute, modulated by some step size to enable gradual maximization towards a desired non-binary attribute intensity. In light of these remarks, the core contribution of this work in terms of novelty is the proposed replacement of traditional supervised approaches  for conditioning with established RL methods represents an interesting, albeit modest innovation.

Evaluation of Style vs Quality Trade-off: The main concern with the proposed approach and the evaluation conducted by the authors is the absence of explicit controls to ensure attribute conditioning does not disrupt generation quality. This is commonly done in conditioning approaches, e.g. by using a reference model for PPO or incorporating non-conditional LM probability in the loss. In the proposed formulation, I do not see any component accounting for overall quality beyond attribute conditioning. In the evaluation, the only metric that does not concern attribute accuracy is diversity measured by Self-BLEU, which is however not sufficient to account for generation quality.

Clarity issues: Despite the stated ability to condition on multiple attributes at once, the examples provided in Appendix G are, in essence, simply showing an increase/decrease in verbosity, and are for this reason not very convincing in showcasing multi-attribute conditioning. Relatedly, while the paper demonstrates strong average performance, it lacks discussion of when and why the method might fail. Analysis of cases where Pre-Control fails to reach target intensities, or where the generated text quality degrades despite matching target scores, would provide valuable insights.

Significance of the results: Regarding the Pareto frontier approximation experiments, the shift in reward resulting from conditioning shown in Figure 4 is barely noticeable, putting in question the significance of the results. In general, the choice of evaluation metrics favor direct optimization of precise attribute values, for which is not surprising that an ad-hoc optimized model achieves the best scores compared to the other tested methods.

[1] Dathathri et al. 2020. Plug and Play Language Models: A Simple Approach to Controlled Text Generation. https://arxiv.org/abs/1912.02164

**Questions:**

Questions:

- Could the authors provide a more detailed breakdown of the computational overhead during inference? Specifically, how does the per-token intervention time scale with sequence length and batch size?
- While the final layer intervention is justified by citation, have the authors empirically compared interventions at different layers or combinations of layers? This ablation would strengthen the architectural choices.
- How does PRE-CONTROL compare to simply fine-tuning models on data filtered to specific attribute ranges? This baseline would help isolate the benefits of test-time intervention.

Minor corrections:

- The font size for Section 3 changes mid-way through the tile. While this was probably done to make the paper fit the page limit, the section title should be consistent in size.

---

> ### Author Response · Authors · 2025-11-23
>
> We are encouraged that you recognized the originality of applying TD learning for this problem, the clarity of our mathematical formulations, and the significance of our method for efficient model adaptation. We provide the following responses to address your concerns and answer your questions:
>
> > The proposed method closely resembles to the established Plug-and-play...
>
> 1. **PRE-CONTROL differs from PPLM in where gradients are computed, which makes our test-time intervention substantially more efficient.**
>
> Although both approaches modify internal activations at inference time, the actual optimization loop is quite different. In PPLM, the attribute models operate on the LM's logits/token distributions, and each intervention step requires: (i) a forward pass through the full language model to obtain logits, (ii) applying the attribute model and computing its loss, (iii) a backward pass to update hidden states. This repeated cycle going through heavy LM head at every decoding step makes PPLM computationally heavy for large LMs and has been noted in later work as a key limitation [5,6,7].
>
> In contrast, PRE-CONTROL learns a lightweight value function directly on hidden states and applies gradients in hidden space, so the LM head does **not** sit inside the optimization loop. As a result, we avoid going through this LM head, leading to a smaller inference-time overhead while still enabling precise, test-time control.
>
> 2. **TD learning V.S. Supervised Fine-tuning**
>
> A natural baseline is to simply label every hidden state with the final sequence-level attribute score and train a supervised regressor. However, recent work [8,9] on reward-guided generation shows that this is **not** an appropriate way to supervise intermediate states. Reward models trained only on full sequences can provide **misleading or even adversarial** guidance when applied to partial sequences, because prefixes with very different futures receive the same terminal reward. This is essentially a credit-assignment failure: all partial states are forced to mimic the final outcome, instead of reflecting their expected eventual attribute intensity. In contrast, TD-style value learning is explicitly designed to address this by propagating the final score back along the trajectory and training the value function to estimate the return from each partial state. This is what makes our per-step estimates suitable for precise, stepwise control during decoding.
>
> Instead of applying SFT for value function training, we conduct experiments to compare PRE-CONTROL to another multi-objective SFT baseline, RiC [4], which represents the non-RL alternative for conditioning on multiple attributes. To keep the rebuttal concise, we refer the reveiwer to our detailed response to the question of supervised fine-tuning baseline below (the last question). The results show that PRE-CONTROL offers better target-matching and Pareto frontier quality with much less computational costs.
>
> 3. **Precise scalar control is not a trivial “maximization with a step size” problem; open-direction methods impose a heavy and impractical tuning burden.**
>
> We appreciate the reviewer’s intuition that one might approximate scalar control by applying an open-direction intervention (e.g., RE-Control or PPLM) under certain amount of steps. However, in practice, this is exactly where open-direction methods break down. They are designed to monotonically increase (or decrease) an attribute, but provide no built-in mechanism for determining when to stop. Achieving “helpfulness = 3” instead of “as helpful as possible” requires knowing, per sample, how many gradient steps to apply. As shown in RE-Control [1] Figure 5b, performance is sensitive to the number of intervention steps. This means that using such open-direction methods for precise intensity control would require: (i) extensive hyper-parameter sweeps over number of steps, and (ii) varied settings for different targets or datasets, since there is no feedback signal to indicate when the desired scalar level is reached.
>
> In contrast, PRE-CONTROL explicitly addresses this issue via two coupled components:(i) a **target-reaching** objective: during intervention, we minimize a MSE loss between the value function’s prediction and the user-specified scalar target and stop once the prediction falls within a small tolerance band. (ii) a **value function** trained on hidden states with TD-$\lambda$ over full decoding trajectories: at each decoding step, it learns to faithfully estimate the final attribute intensity from the current hidden state. Together, these give a principled stopping rule and removes the need for exhaustive hyper-parameter tuning over step counts. In multi-attribute settings, this becomes even more critical: manually tuning separate step sizes for each attribute and each target vector quickly becomes infeasible, whereas PRE-CONTROL directly optimizes the multi-dimensional deviation to the target vector in a unified objective.

---

> > ### Author Response · Authors · 2025-11-23
> >
> > > The main concern with the proposed approach...
> >
> > Since PRE-CONTROL is a test-time intervention method, it does not maintain a separate reference policy as in PPO-style RLHF, where a KL term to the reference model explicitly regularizes generation. Our method instead edits hidden representations at test time, rather than optimizing a new token-level objective. For this reason, we do not include an explicit KL-divergence or base-LM log-probability term. Incorporating such a loss would require mapping each edited representation back through the large LM head to obtain a full token distribution at every intervention step, which would substantially increase computational overhead and defeat the purpose of a lightweight, test-time control method.
> >
> > To further evaluate the generation quality, we adopt **Coherence** which calculates the cosine similarity between the embeddings of the prompt and its continuation. We follow [1,2] to use the pre-trained SimCSE sentence embedding model to obtain these embeddings. We evaluate the repsonses generated by Phi-4 on Code-UltraFeedback dataset. The results in the tables below show that PRE-CONTROL maintains coherence on par with the base model and other test-time control baselines. This indicates that our interventions do not degrade overall generation quality while providing much more precise control over attribute targets.
> >
> > ### Relative Positive Representative Target Score
> > |Method|Coherence|
> > |-|-|
> > |Base|0.7744|
> > |Prompting | 0.7746|
> > |ITI| **0.7752**|
> > | RE-Control|0.7742|
> > |MAT-Steer|0.7748|
> > |PRE-CONTROL| 0.7747|
> >
> > ### Relative Negative Representative Target Score
> > |Method|Coherence|
> > |-|-|
> > |Base|0.7744|
> > |Prompting|0.7746|
> > |ITI|0.7751|
> > |RE-Control|0.7744|
> > |MAT-Steer|**0.7772**|
> > |PRE-CONTROL|0.7746|
> >
> > > Despite the stated ability to condition on...
> >
> > Case Study: We would like to clarify that the qualitative examples in Appendix G are not single-attribute (verbosity-only) cases. In the negative target example, the intervention shifts the score from [4,4,4,3,3] to [3,3,3,2,2], with all five attributes reduced. In the positive target example, the intervention moves towards a more complex setting, from [4,4,4,1,1] to [4,4,4,2,2], increasing both complexity and verbosity while keeping the other attributes high. In the revised manuscript, we include **additional qualitative example** in Appendix G to more clearly illustrate the effectiveness of our method (with only changing one attribute).
> >
> > Failure Analysis: We agree that it is important to understand when and why PRE-CONTROL may fail. we observe that PRE-CONTROL is less effective on **targets that are rarely observed in the underlying data distribution**. For example, for the extreme target [4, 4, 4, 3, 3] in Table 11, the absolute success rate is relatively low ($\approx$ 1.8%). This primarily stems from the **distributional coverage of the Value Function training data**. As detailed in Appendix D.3, $V_{\phi}$ is trained on samples generated by the base model itself. Consequently, attribute combinations that lie far outside the base model’s natural distribution receive **sparse supervision** during training, which can lead to less accurate value estimates and weaker gradient guidance at inference time. In these out-of-distribution regions, PRE-CONTROL may therefore fail to reach the exact target intensities, even though it still tends to push the generation in the desired direction.
> >
> > > Regarding the Pareto frontier approximation...
> >
> > For Coherence reward, there are **21%** samples scored in [4.0, 5.0] before the intervention; after intervention, the samples scored in [4.0, 5.0] are boosted to **28%**. For Complexity, there are **44%** samples scored in [2.0, 3.0] before the intervention; after intervention, the samples are boosted to **48%**. To facilitate interpreting Fig. 4, we provide the mean reward scores for these two attributes in the table below. The average of both reward scores increases, validating the effectiveness of our method. Besides Fig. 4, we also present a direct visual comparison of Pareto frontier in Fig 3. and quantify Pareto quality with frontier-level metrics (HyperVolume & Sparsity following [3]) in Table 2. Our method demonstrates strong performance under all these evaluations.
> >
> > Regarding the choice of general evaluation metrics, we would like to clarify that one of our core contribution is precisely to reformulate alignment as an attribute target-reaching problem (See Introduction). Metrics such as Success Rate and Distance-to-Target are natural for this problem and are **not** specific to our architecture.
> >
> > | Coherence | Mean Reward Score |
> > |---|---|
> > | Before| 3.72 |
> > | After| **3.95** |
> >
> > |Complexity| Mean Reward Score |
> > |-|-|
> > |Before| 2.04      |
> > | After     | **2.11**      |

---

> > > ### Author Response · Authors · 2025-11-23
> > >
> > > ## Questions
> > > > Could the authors provide a more...
> > >
> > > We provide the detailed breakdown of computational cost by varying batch size and output token length in the table below. We conduct experiments on HelpSteer2 by Llama-3.2-3B. For batch size experiment, we fix the output token length to be 512. For output token length experiment, we fix the batch size to be 24.
> > >
> > > | Output Token Length | GPU Hours |
> > > |--|-|
> > > | 256| 0.04|
> > > | 512| 0.09      |
> > > | 768          | 0.14     |
> > >
> > > | Batch Size | GPU Hours |
> > > |---------------|-----------|
> > > | 12          | 0.14     |
> > > | 24    | 0.09      |
> > > | 48     | 0.05      |
> > >
> > > > While the final layer intervention is justified by citation...
> > >
> > > Thank you for your interest in the empirical comparison of the interventions at different layers. Architecturally, PRE-CONTROL is agnostic to the layer index: as long as the value function $V_\phi$ (Eq. 4) is trained on the hidden states from a given layer, the same target-reaching objective and gradient-based intervention (Eq. 8) apply.
> > >
> > > That said, for any intermediate layer $\ell$, we could instead train a value function $V_\phi^{(\ell)}$ on $h_t^{(\ell)}$ and apply the same update there, propagating the modified representation through layers $\ell+1, \dots, L$ before computing the logits. We chose to instantiate the method on the final layer primarily because (i) prior work suggests that semantic and task-specific information is most linearly decodable at upper layers, which makes attribute intensities easier to predict, and (ii) intervening one step before the logits yields more local and stable changes to the token distribution, with less interference with lower-level syntactic or factual representations.
> > >
> > > A systematic ablation over intervention layers is, however, not “for free”: for each candidate layer $\ell$, we need to re-train a separate value function $V_\phi^{(\ell)}$ on that layer’s representations and re-run the full generation and evaluation pipeline. During the discussion phase so far, most of our computation and time were devoted to (1) scaling to larger models, (2) hyperparameter studies for intervention dynamics, and (3) generation-quality evaluations. As a result, we have not yet completed this architectural ablation.
> > >
> > > If the reviewer and AC consider this architectural ablation critical, we are happy to prioritize training a value function on a representative middle layer and report the corresponding results later in the rebuttal period, subject to time and compute constraints.

---

> ### Author Response · Authors · 2025-11-23
>
> > How does PRE-CONTROL compare to simply fine-tuning models...
>
> We would like to clarify that fine-tuning on data filtered to a specific attribute band addresses a fundamentally different use case from PRE-CONTROL. Our goal is to enable **flexible, test-time control**: a single base model can be steered to **arbitrary scalar targets** without any further training. In contrast, filtering the training data to a narrow range (e.g., “high helpfulness, high harmlessness, medium conciseness”) and fine-tuning would produce a specialized policy that is effective only for that particular region of attribute space. Changing the target or exploring the Pareto frontier would require (i) constructing a new filtered dataset for each band or preference vector and (ii) running another full fine-tuning pipeline, which quickly becomes impractical as the number of target configurations grows.
>
> To directly connect with the reviewer’s suggestion, we **include a SFT–based multi-objective alignment baseline: Rewards-in-Context (RiC)** [4]. RiC first labels each prompt-response pair with a reward vector $r(x,y)=[r_1(x,y), \dots,r_N(x,y)]$, and then append such a reward vector to the prompt in a structured textual form (e.g. special token $\langle R_1\rangle r_1,\dots,\langle R_N\rangle r_N$. The policy $\pi$ is **supervised-finetuned to imitate $y$ given the augmented input $(x,r(x,y))$**. This teaches the model to ground its response in the reward vector. RiC further augments the dataset with responses that lie near the empirical Pareto front (via multi-objective rejection sampling) and applies the same **reward-conditional SFT loss** to refine the policy. During inference, user preferences are mapped to desired reward vectors, which are injected into the prompt, and the fine-tuned model generates responses accordingly. This approach is better aligned with our problem than training separate models per reward band: a **single conditional SFT model** learns to realize different reward configurations through conditioning.
>
> We mirror our main experiments setting to compare PRE-CONTROL with RiC on HelpSteer2 by Llama-3.2-3B. We would like to emphasize that PRE-CONTROL is a more efficient test-time method, different from RiC which requires multi-stage supervised fine-tuning. RiC's training + inference time costs approximately 9.8 GPU hours, whereas PRE-CONTROL's value function training + test-time intervention consumes about 0.6 GPU hours, which is 16x faster. In addition to such efficiency, the results shown in the table below demonstrate that PRE-CONTROL achieves stronger performance than RiC.
>
> | Method        | Diversity | $l_1$ Distance | Success Rate(%) | HyperVolume
> |---------------|-----------| -----------| -----------| -----------|
> | RiC    | 0.683      | 4.08 | 7.49 | 8.28
> | PRE-CONTROL     | **0.529**  | **2.65** | **7.73** | **11.08**
>
> > The font size for Section 3 changes mid-way through...
>
> Thanks for your carefully reading. We have changed it to consistent font size in the revised manuscript.
>
> [1] Kong Lingkai, et al. Aligning Large Language Models with Representation Editing: A Control Perspective, 2024. NeurIPS
>
> [2] Su Yixuan, et al. A Contrastive Framework for Neural Text Generation, 2022. NeurIPS
>
> [3] Zhong Yifan, et al. Panacea: Pareto Alignment via Preference Adaptation for LLMs, 2024. NeurIPS
>
> [4] Yang Rui, et al. Rewards-in-Context: Multi-objective Alignment of Foundation Models with Dynamic Preference Adjustment, 2024. ICML
>
> [5] Krause Ben, et al. GeDi: Generative Discriminator Guided Sequence Generation, 2021. EMNLP
>
> [6] Chen et al. FAST: Improving Controllability for Text Generation with Feedback-Aware Self-Training, 2022. EMNLP
>
> [7] Madotto et al. Plug-and-Play Conversational Models, 2020 EMNLP
>
> [8] Lee Youngwon, et al. Learning to Rank Generation with Pairwise Partial Rewards, 2023. EMNLP
>
> [9] Rashid Ahmad, et al. A Critical Look At Tokenwise Reward-Guided Text Generation, 2025. COLM
>
> ---
> We sincerely appreciate your thoughtful and constructive review. Your comments helped us better articulate the goals, scope, and implications of our work. We have incorporated additional qualitative examples in the revised manuscript. We hope that our clarifications and detailed responses can addressed your concerns and convince you to lean more towards acceptance of our paper.

---

### Official Review · Reviewer_bFK2 · 2025-10-28

**Soundness:** 2
**Presentation:** 3
**Contribution:** 2
**Rating:** 4
**Confidence:** 3

**Summary:**

This paper proposes a targeted representation editing method for precise attribution intensity control. It formulates precise attribute intensity control as a target-reaching problem, trains a lightweight value function and  performs representation intervention to control LLM outputs. Extensive experiments are conducted to verify the effectiveness of the proposed method.

**Strengths:**

1.The research topic is both timely and interesting. Fine-grained control over LLM behavior is an emerging and essential direction for practical LLM applications.

2.The explanation of the method, including the mathematical formulations and step-by-step process, is clear and easy to follow.

3.The discussion and visualization of the efficient Pareto frontier approximation are well-presented. This component effectively addresses the challenge of handling multiple conflicting attributes.

**Weaknesses:**

1.During the test phase, the proposed method appears to require iterative optimization at each token generation step to modify hidden states (as shown in Eq. 8). This could introduce substantial computational overhead. What is the average response generation time compared to the base LLM and other baselines?

2.The scalability of the method should be further validated, especially on larger models (e.g., 13B or 30B parameters).

3.It is unclear how the scores of generated responses in the experiments are obtained. Are they scored using the ArmoRM reward model? If so, please clarify the evaluation setup and justify whether the LLM-generated scores provides accurate and reliable assessments.

4.For text generation tasks, human evaluation should be included as a complementary measure, as automatic metrics and reward models may exhibit bias and fail to capture nuanced quality differences.

5.The sensitivity of the hyperparameters in Eq. 9 should be analyzed.

**Questions:**

1.How is $w_i$ in Eq. 9 determined？

2.Could you clarify where the original attribute score in the datasets and the reward model score are used in your study? It is somewhat unclear how these two scores interact or are combined during training or inference.

**Details Of Ethics Concerns:**

N.A.

---

> ### Author Response · Authors · 2025-11-23
>
> We appreciate your positive feedback on the timeliness of our research topic and the clarity of our method's explanation. We are also glad you found our discussion of the Pareto frontier effective. Below, we address your concerns and questions:
>
> > 1.During the test phase, the proposed method appears to require iterative optimization at each token generation step to modify hidden states (as shown in Eq. 8). This could introduce substantial computational overhead. What is the average response generation time compared to the base LLM and other baselines?
>
> We report the computational cost for generating 434 test samples of HelpSteer2 using Llama-3.2-3B in the table below (measured on 4xH200 GPUs). While PRE-CONTROL incurs moderate overhead due to token-level optimization, it remains within the same order of magnitude as other baselines. Given its strong performance in multi-attribute controllability, we believe this overhead is a practical trade-off.
>
>
> | Method        | GPU Hours |
> |----------|-----------|
> | Base Model    | 0.02      |
> | Prompting     | 0.02      |
> | ITI   | 0.06      |
> | MAT-Steer     | 0.07   |
> | RE-Control*   | 0.10   |
> | PRE-CONTROL   | 0.09 |
>
> \* RE-Control's computational cost is sensitive to the number of intervention steps. We follow [9] to use 100 steps which yields the best performance in terms of reward scores.
>
> > 2.The scalability of the method should be further validated, especially on larger models (e.g., 13B or 30B parameters).
>
> We agree that validation on larger models is important. In the revised draft, we have added experiments on **Phi-4-14B** to assess scalability. The setup (datasets, target configurations, evaluation metrics) mirrors our main experiments. As shown in the table below, across both HelpSteer2 and Code-UltraFeedback and for both positive and negative representative targets, PRE-CONTROL consistently attains (i) the **highest success rate**, and (ii) the **best diversity** (the lower the better), while maintaining competitive or better L1 distance compared to all baselines (Base, Prompting, ITI, Re-Control, MAT-Steer). These new results have been updated in **Table 1** in the revised draft, and they support that our method scales to substantially larger models without architectural changes.
>
> ### Relative Positive Representative Target Score
>
> | Method     | HelpSteer2 (Div; L1; SR) [4,4,4,2,2] | Code-UltraFeedback (Div; L1; SR) [3,3,3,3,3] |
> |--|---|-------|
> | Base      | 0.918; 2.04; N/A        | 0.979; 0.694; N/A    |
> | Prompting | 0.946; 2.01; 1.93       | 0.987; 0.632; 10.34  |
> | ITI       | 0.709; 2.04; 5.06       | 0.920; 0.584; 16.33   |
> | Re-Control | 0.914; 1.99; 3.61       | 0.972; 0.656; 11.29  |
> | MAT-Steer | 0.697; 2.29; 3.86       | 0.912; 0.569; 21.66 |
> | **Ours**  | **0.686; 1.93; 7.23**   | **0.880; 0.560; 26.96**  |
>
> ### Relative Negative Representative Target Score
>
> | Method     | HelpSteer2 (Div; L1; SR) [3,3,3,2,2] | Code-UltraFeedback (Div; L1; SR) [2,2,2,2,2] |
> |-----|----|-----|
> | Base      | 0.930; 2.79; N/A        | 0.963; 4.58; N/A  |
> | Prompting | 0.962; 2.78; 1.25       | 0.974; 4.57; 0.31   |
> | ITI    | 0.656; 2.83; 2.02  | 0.778; 4.62; 1.65 |
> | Re-Control | 0.888; 2.82; 2.49       | 0.917; 4.56; 0.72  |
> | MAT-Steer | 0.644; **2.72**; 2.73       | 0.651; 4.59; 1.65    |
> | **Ours**  | **0.602**; 2.83; **3.74**   | **0.605**; **2.89**; **19.14** |
>
> > It is unclear how the scores of generated responses in the experiments are obtained. Are they scored using the ArmoRM reward model? If so, please clarify the evaluation setup and justify whether the LLM-generated scores provides accurate and reliable assessments.
>
> As described in line 907-910, all generated responses are evaluated using the ArmoRM-Llama3-8B reward model [1]. For each prompt–response pair, ArmoRM outputs a 5-dimensional real-valued vector of attribute scores, which we then use for both training the value function and computing our evaluation metrics (Success Rate, Distance to Target).
>
> We would like to clarify that the reward model used in our paper **does provide accurate and reliable assessments**. ArmoRM-Llama3-8B is specifically trained on large corpus of multi-attribute human preference data, encompassing 8 datasets with approximate 585.4K samples. Its multi-objective architecture and Mixture-of-Experts gating can automatically selects and scalarizes relevant objectives based on the context, thus capture nuanced trade-offs between alignment dimensions faithfully. As a result, it achieves SOTA performance on RewardBench [2] (a comprehensive evaluation set for reward models), surpassing GPT-4-as-a-judge and approaching the performance of much larger reward models. Beyond its own paper, ArmoRM has been adopted as a reward model/evaluator by other works [3,4,5]. More importantly, both of our evaluation datasets lie in its training corpus. Together, these factors make ArmoRM a reliable, human-grounded proxy for evaluating attribute-level control on our benchmark datasets.

---

> > ### Author Response · Authors · 2025-11-23
> >
> > > For text generation tasks, human evaluation should be included as a complementary measure, as automatic metrics and reward models may exhibit bias and fail to capture nuanced quality differences.
> >
> > 1. **Clarifying Scope**: We respectfully clarify that the primary research question of this work is Precise Test-Time Control: Given a defined reward signal, how accurately can we steer the model to hit a specific scalar value? Therefore, using the same high-fidelity Reward Model (ArmoRM) for both supervision and evaluation is mathematically necessary to isolate the performance of the steering mechanism from the noise of the evaluator.
> > 2. **Reliability of ArmoRM**: We acknowledge the concern regarding automatic metrics. However, we selected ArmoRM-Llama3-8B specifically because it currently represents the state-of-the-art in open-weights reward modeling, serving as a highly reliable proxy for human preference. We support this with two key comparisons:
> >
> >     - **GPT-4 as a Human Proxy**: Recent literature establishes that strong LLMs like GPT-4 achieve high agreement with human experts. For instance, Zheng et al. [6] report that GPT-4 achieves over **80% agreement** with humans, comparable to human-human inter-annotator agreement.
> >     - **ArmoRM vs. GPT-4**: Crucially, ArmoRM-Llama3-8B outperforms GPT-4 on rigorous benchmarks. On RewardBench [2], ArmoRM achieves a score of **89.0**, surpassing **GPT-4 Turbo (84.3)** by a significant margin. This superior performance stems from its large-scale and high-quality training data and its **Multi-Objective Mixture-of-Experts** as mentioned in the above response.
> >
> > 3. **Domain Consistency**: Furthermore, ArmoRM is trained on a massive corpus (585K samples) that explicitly includes the HelpSteer2 dataset used in our experiments. This ensures that the reward signal is not out-of-distribution but rather a "ground truth" representation of the attribute definitions (Complexity, Verbosity, etc.) inherent to the dataset.
> >
> > In conclusion, given that ArmoRM surpasses GPT-4 (a widely accepted human proxy) and is domain-aligned with our test set, we believe it provides a stricter and more consistent signal for measuring "control precision" than small-scale human evaluation, which often suffers from high variance on scalar intensity tasks.
> >
> > > The sensitivity of the hyperparameters in Eq. 9 should be analyzed.
> >
> > We conduct the sensitivity analysis on intervention step size $\alpha$. The experiment As shown in the table below, PRE-CONTROL's controllability remains stable (between 6.51% and 6.99%) across a wide range of $\alpha$ values. This confirms the stability of our method.
> > | Step Size        | Success Rate |
> > |---------------|-----------|
> > | 1e-1    | 6.51      |
> > | 5e-2     | 6.99      |
> > | 1e-2           | 6.51     |
> > | 5e-3     | 6.51      |
> > | 1e-3    | 6.99      |
> >
> > ## Questions
> > > How is $w_i$ in Eq. 9 determined？
> >
> > $w_i$ is the user-specified relative importance of each attribute. Empirically, we treat each attribute equally important, set $w_i=1$ (See line 911).
> >
> > > Could you clarify where the original attribute score in the datasets and the reward model score are used in your study? It is somewhat unclear how these two scores interact or are combined during training or inference.
> >
> > We do not use the original dataset attribute scores in our method. Both training (value function learning) and evaluation rely exclusively on the ArmoRM reward model's predictions. This design choice ensures consistency: since ArmoRM is specifically trained on HelpSteer2 and Code-UltraFeedback (the datasets we evaluate on), its scores provide a reliable signal throughout our pipeline.
> >
> > [1] Wang, Haoxiang, et al. "Interpretable preferences via multi-objective reward modeling and mixture-of-experts, 2024b." EMNLP
> >
> > [2] Lambert Nathan, et al. RewardBench: Evaluating Reward Models for Language Modeling, 2025. NAACL
> >
> > [3] Razin Noam, et al. What Makes a Reward Model a Good Teacher? An Optimization Perspective, 2025. NeurIPS
> >
> > [4] Deshpande Vijeta, et al. Diverse, not Short: A Length-Controlled Data Selection Strategy for Improving Response Diversity of Language Models, 2025. EMNLP
> >
> > [5] Bai Chenjia, et al. Online Preference Alignment for Language Models via Count-based Exploration, 2025. ICLR
> >
> > [6] Zheng et al., "Judging LLM-as-a-Judge with MT-Bench and Chatbot Arena", NeurIPS 2024.
> >
> > ---
> > We sincerely appreciate the reviewer’s constructive feedback. We have carefully addressed all of the raised questions and uploaded a revised version of the paper that incorporates the new experiments discussed above, with major changes clearly highlighted. We believe these revisions have further strengthened our work, and we hope our responses satisfactorily resolve your concerns and encourage you to view the paper more favorably for acceptance.

---

### Official Review · Reviewer_VUE9 · 2025-11-01

**Soundness:** 2
**Presentation:** 3
**Contribution:** 2
**Rating:** 4
**Confidence:** 4

**Summary:**

This paper presents a relatively precise solution for multi-attribute control through test-time intervention. The algorithm is validated on two common tasks—text generation and code generation—demonstrating its feasibility.

**Strengths:**

1. The paper introduces the $TD(\lambda)$ algorithm to assess partial generation outputs, thereby resolving the dependency on full generation cycles typical of conventional approaches.

2. It employs a test-time gradient descent strategy to approximate target attribute scores. This is coupled with an efficient Pareto frontier approximation technique that boosts efficiency, eliminates the need for extensive retraining, and ensures broad adaptability.

3. Through comprehensive empirical evaluations on a variety of tasks against multiple baselines, the study validates the algorithm's significant improvements in both precision and computational efficiency.

**Weaknesses:**

1. The novelty of the work is modest. The proposed test-time intervention (Eq. 8) shares its core principle (gradient descent) with existing methods like Cold Decoding [1] and BOLT [2].

2. What is the relationship between $v_{\phi}(h_{t})$ and  $v_{\phi}(s_{t})$ in Equations 4, 5, and 6? Are they equivalent?

3. The experimental validation is based on 3B models. It would strengthen the work to demonstrate the method's applicability on larger-scale models.

4. Regarding the diversity metric in Table 1: For an attribute control task, why is higher diversity considered better? How is "Distance to Target" precisely calculated? Is it derived from the attribute score of the generated text? If so, please specify the method for calculating this generated text score. For "Success Rate", how is $N_{aligned}$ defined? Is a sample considered aligned only if its score exactly matches the target score across all five dimensions?

5. Are the predefined target scores (e.g., [4,4,4,2,2]) in Table 1 chosen randomly? Would the method yield similar improvements for other arbitrary target score combinations?

[1]COLD Decoding: Energy-based Constrained Text Generation with Langevin Dynamics
[2]BOLT: Fast Energy-based Controlled Text Generation with Tunable Biases

**Questions:**

See the weakness.

---

> ### Author Response · Authors · 2025-11-23
>
> Thank you for your detailed and thoughtful review. We appreciate your positive comments on our method's efficiency and adaptability, and the breadth of our empirical evaluation. Below, we address your concerns and questions point by point:
>
> > The novelty of the work is modest. The proposed test-time intervention (Eq. 8) shares its core principle (gradient descent) with existing methods like Cold Decoding [1] and BOLT [2].
>
> We thank the reviewer for the comparison to COLD Decoding and BOLT. We respectfully disagree with the assessment that the novelty is modest due to the shared use of gradient descent. While it is true that all three methods utilize gradient-based updates, this represents a shared mathematical operation, not a shared algorithmic paradigm. Asserting they are identical because they use gradients is akin to claiming training a neural network and performing linear regression are the same because both use SGD. We highlight the fundamental divergences below:
> ## 1. Distinct Optimization Objectives:
>
> - **COLD Decoding & BOLT** are rooted in Energy-Based Models (EBMs). Their objective is to minimize energy (maximize reward), driving the generation toward the extreme of an attribute scale (e.g., "make this text as positive as possible").
>
> - **PRE-CONTROL Innovation**: Our method reformulates alignment as a **Target-Reaching Problem** (Eq. 2). We minimize the squared error $(V_\phi - \tau)^2$ between a predicted value and a specific scalar target. This creates a convex optimization landscape centered on a setpoint (e.g., "helpfulness = 4"). This enables precise, fine-grained control that maximization-based EBMs cannot naturally achieve without significant re-engineering.
>
> ## 2. Distinct Intervention Mechanisms:
>
> The reviewer notes the similarity in the update rule (Eq. 8), but the critical differentiator is the source of the gradient signal and the temporal dynamics of the intervention.
>
> - **COLD Decoding & BOLT (Iterative Search)**: These methods rely on a static energy function $E$. At test time, they must perform an iterative search (Langevin dynamics or gradient tuning) at the soft toekn level or the output logits level to locate a low-energy sequence. This approach is computationally expensive because its lack of intermediate feedback. For example, the backpropagation in BOLT only happens after the whole sequence is generated, which means there is no guidance during the decoding process, so there is a high latency between error and correction.
> - **PRE-CONTROL (Look-ahead Control)**: We utilize a **Learned Value Function** ($V_\phi$) trained via TD learning. This function acts as a look-ahead mechanism, predicting the future outcome from the current state. Eq. 8 applies a direct, calculated correction based on this prediction. This enables **real-time, per-token intervention** without the expensive inner-loop search required by COLD and BOLT.
>
>
> In conclusion, PRE-CONTROL solves a different problem (Precise Scalar Control) using a different mechanism (Value-Function Guided Steering) than the baselines. The shared use of gradient descent is merely the mechanism of update, but the "brain" driving that update ($V_\phi$ vs. Energy) and the goal it drives toward ($\tau$-target vs. Max Reward) are fundamentally novel.

---

> ### Author Response · Authors · 2025-11-23
>
> > The experiments only compared small-scale LLMs; validation with larger models such as 7B or 8B parameters would be more convincing and strengthen the findings.
>
> We agree that validation with larger models would strengthen our findings. In the revised draft, we have added experiments on **Phi-4-14B** to assess scalability. The setup (datasets, target configurations, evaluation metrics) mirrors our main experiments. As shown in the table below, across both HelpSteer2 and Code-UltraFeedback and for both positive and negative representative targets, PRE-CONTROL consistently attains (i) the **highest success rate**, and (ii) the **best diversity** (the lower the better), while maintaining competitive or better L1 distance compared to all baselines. These new results have been updated in **Table 1** in the revised draft, and they support that our method scales to substantially larger models without architectural changes.
>
>
> ### Relative Positive Representative Target Score
>
> | Method     | HelpSteer2 (Div; L1; SR) [4,4,4,2,2] | Code-UltraFeedback (Div; L1; SR) [3,3,3,3,3] |
> |-----------|--------------------------|----------------------------------|
> | Base      | 0.918; 2.04; N/A        | 0.979; 0.694; N/A                |
> | Prompting | 0.946; 2.01; 1.93       | 0.987; 0.632; 10.34              |
> | ITI       | 0.709; 2.04; 5.06       | 0.920; 0.584; 16.33              |
> | Re-Control | 0.914; 1.99; 3.61       | 0.972; 0.656; 11.29              |
> | MAT-Steer | 0.697; 2.29; 3.86       | 0.912; 0.569; 21.66              |
> | **Ours**  | **0.686; 1.93; 7.23**   | **0.880; 0.560; 26.96**          |
>
>
> ### Relative Negative Representative Target Score
>
> | Method     | HelpSteer2 (Div; L1; SR) [3,3,3,2,2] | Code-UltraFeedback (Div; L1; SR) [2,2,2,2,2] |
> |-----------|--------------------------|----------------------------------|
> | Base      | 0.930; 2.79; N/A        | 0.963; 4.58; N/A                |
> | Prompting | 0.962; 2.78; 1.25       | 0.974; 4.57; 0.31               |
> | ITI       | 0.656; 2.83; 2.02       | 0.778; 4.62; 1.65               |
> | Re-Control | 0.888; 2.82; 2.49       | 0.917; 4.56; 0.72               |
> | MAT-Steer | 0.644; **2.72**; 2.73       | 0.651; 4.59; 1.65               |
> | **Ours**  | **0.602**; 2.83; **3.74**   | **0.605**; **2.89**; **19.14**           |

---

> > ### Author Response · Authors · 2025-11-23
> >
> > ## Questions
> > > What is the relationship between $v_{\phi}(h_t)$ and $v_{\phi}(s_t)$ in Equations 4, 5, and 6? Are they equivalent?
> >
> > Thank you for pointing out this. In our implementation, the “state” at step t is fully represented by the hidden state $h_t$ of the backbone LLM (i.e., the partial context and generation up to step $t$). From Eq. 4 to Eq. 5, we switch $v_\phi(h_t)$ to the standard TD($\lambda$) notation $v_\phi(s_t)$  to align with the RL literature. In the revised manuscript, we have made this explicit by stating that $s_t$​ is represented by $h_t$​ to avoid confusion (see line 155). This is purely notational and does not affect the algorithm or experiments.
> >
> >
> > > Regarding the diversity metric in Table 1: For an attribute control task, why is higher diversity considered better?
> >
> > We aim to maintain high diversity while hitting the desired attribute intensities because we do not want to collapse models' generative behavior into generic, repetitive outputs. It has been demonstrated that many methods trade off diversity for stronger control, as prior works [1,2] has shown that aggressively optimizing for constraints tends to produce degenerate responses that are fluent but generic and low-diversity. Therefore, a higher diversity score indicates that the model does not keep parroting similar outputs after steering.
> >
> > > How is "Distance to Target" precisely calculated? Is it derived from the attribute score of the generated text? If so, please specify the method for calculating this generated text score.
> >
> > The "Distance to Target" is computed as $l_1$ distance between the attribute scores of the generated text and the target score, $\sum^{5}_{i=1}|\hat{R}_i - \tau_i|$. As described in line 854-856, we use ArmoRM-Llama-8B reward model to evaluate each generated text to output a real value 5-dimensional vector of attribute scores.
> >
> > > For "Success Rate", how is $N_{aligned}$ defined? Is a sample considered aligned only if its score exactly matches the target score across all five dimensions?
> >
> > In the evaluation for "Success Rate", we define $N_{aligned}$ as samples whose ArmoRM-Llama-8B scores, after rounding each of the five dimensions to the nearest integer, exactly match target scores in all dimensions. We further clarify this in the updated draft (See line 287-290).

---

> > > ### Author Response · Authors · 2025-11-23
> > >
> > > > Are the predefined target scores (e.g., [4,4,4,2,2]) in Table 1 chosen randomly? Would the method yield similar improvements for other arbitrary target score combinations?
> > >
> > > 1. **Are the scores in Table 1 random?** No, the scores in Table 1 (e.g., [4,4,4,2,2]) were **not chosen randomly**. They are explicitly defined as "representative target scores" based on the statistical distribution of attribute combinations naturally found in the dataset.
> > >     - Our paper states in the caption of Table 1 that these targets are defined based on the "statistical distribution of attributes combination in each dataset".
> > >     - Figure 5 visualizes these distributions, showing that specific combinations (like [4,4,4,2,2] for HelpSteer2) are representative modes in the data distribution of the base model.
> > > 2. **Does the method work for arbitrary targets?** Yes, the method demonstrates robust performance across a wide range of arbitrary target combinations.
> > >     - The paper addresses this directly in **Appendix C.3**. Specifically, **Tables 11 and 12** in the Appendix provide a "comprehensive evaluation across a wider range of target scores".
> > >     - The results show consistent improvements in Success Rate and Distance to Target for various other combinations, confirming that the method's effectiveness is not limited to the representative targets shown in Table 1.
> > > 3. **More discussion.** We also note that for some **targets that are rarely observed in the data distribution** (e.g., [4, 4, 4, 3, 3] in Table 11), the absolute success rate is relatively lower ($\sim 1.8\%$). We believe this stems from the **distributional coverage of the Value Function training data**. As detailed in Appendix D.3, $V_\phi$ is currently trained on samples generated by the base model itself. Consequently, for attribute combinations that lie significantly outside the base model's natural distribution, the Value Function receives sparse supervision during training, potentially leading to less precise gradient guidance during inference.
> > >
> > >     However, we emphasize the following points:
> > >
> > >     1. **Relative Superiority**: Even in these challenging, out-of-distribution regions, PRE-CONTROL consistently outperforms baselines. For example, for the difficult target [4, 4, 4, 3, 3] on HelpSteer2, PRE-CONTROL achieves a success rate of **1.81%**, nearly double that of Prompting (1.20%) and significantly higher than RE-Control (0.80%).
> > >     2. **Distance Metric**: While the "exact match" (Success Rate) is relatively lower on certain targets, PRE-CONTROL still reduces the $l_1$ Distance to Target (3.02) compared to the Base model (3.12), indicating that the method is successfully "tugging" the generation in the right direction, even if it cannot fully overcome the model's prior to reach the exact target.
> > >     3. Future work could intentionally target such rare combinations through active data collection, exploration strategies, or reweighting schemes during value function training to enhance performance in these practically relevant but underrepresented regions of the attribute space.
> > >
> > > [1] Gu, Yuxuan, et al. "Improving controllable text generation with position-aware weighted decoding." Findings of the Association for Computational Linguistics: ACL 2022. 2022.
> > >
> > > [2] Holtzman, Ari, et al. "The curious case of neural text degeneration." arXiv preprint arXiv:1904.09751 (2019).
> > >
> > > ---
> > > Thank you for your detailed review. We have taken great care in addressing the questions raised. The revised manuscript incorporates all requested clarifications and new experimental results with clear highlighting. We believe these additions substantially strengthen the paper's contribution and clarity, and we hope our responses satisfactorily address your questions and merit your consideration for acceptance.

---

### Official Review · Reviewer_HVF5 · 2025-11-02

**Soundness:** 3
**Presentation:** 3
**Contribution:** 3
**Rating:** 6
**Confidence:** 4

**Summary:**

This paper presents PRE-CONTROL, a method for achieving precise attribute intensity control in large language models through targeted representation editing. The authors reformulate attribute control as a target-reaching problem rather than simple optimization, training a lightweight value function using temporal-difference learning to predict attribute scores from partial generations. During inference, they apply gradient-based interventions on hidden representations to steer model outputs toward user-specified attribute intensities on a continuous scale. Experiments on LLaMA-3.2-3b and Phi-4-mini using HelpSteer2 and Code-UltraFeedback datasets demonstrate superior performance compared to baseline methods in achieving target attribute scores.

**Strengths:**

1. The paper demonstrates innovation by directly editing model features to achieve precise control over output attributes according to user preferences. This represents a notable contribution, as previous methods have not considered direct feature editing for attribute control.
2. The method innovatively predicts expected target scores for entire sentences based on partially generated tokens, which is more efficient than token-by-token prediction approaches.
3. The method demonstrates strong experimental performance.

**Weaknesses:**

1. The experiments only compared small-scale LLMs; validation with larger models such as 7B or 8B parameters would be more convincing and strengthen the findings.

2. Consider incorporating discussions of recent multi-objective alignment literature to provide better context and positioning relative to the current state of the field.

   [1] PARM: Multi-Objective Test-Time Alignment via Preference-Aware Autoregressive Reward Model

   [2] Rewards-in-Context: Multi-objective Alignment of Foundation Models with Dynamic Preference Adjustment

   [3] Controllable Preference Optimization: Toward Controllable Multi-Objective Alignment

   [4] Preference Orchestrator: Prompt-Aware Multi-Objective Alignment for Large Language Models

**Questions:**

see weaknesses.

---

> ### Author Response · Authors · 2025-11-23
>
> We appreciate your recognition of our method's innovation and notable contribution in direct feature editing for attribute control, and its superior experimental performance. We address your concerns below:
>
> > The experiments only compared small-scale LLMs; validation with larger models such as 7B or 8B parameters would be more convincing and strengthen the findings.
>
> We agree that validation on larger models is important. In the revised draft, we have added experiments on **Phi-4-14B** to assess scalability. The setup (datasets, target configurations, evaluation metrics) mirrors our main experiments. As shown in the table below, across both HelpSteer2 and Code-UltraFeedback and for both positive and negative representative targets, PRE-CONTROL consistently attains (i) the **highest success rate**, and (ii) the **best diversity** (the lower the better), while maintaining competitive or better L1 distance compared to all baselines (Base, Prompting, ITI, Re-Control, MAT-Steer). These new results have been updated in **Table 1** in the revised draft, and they support that our method scales to substantially larger models without architectural changes.
>
>
> ### Relative Positive Representative Target Score
>
> | Method     | HelpSteer2 (Div; L1; SR) [4,4,4,2,2] | Code-UltraFeedback (Div; L1; SR) [3,3,3,3,3] |
> |-----------|--------------------------|----------------------------------|
> | Base      | 0.918; 2.04; N/A        | 0.979; 0.694; N/A                |
> | Prompting | 0.946; 2.01; 1.93       | 0.987; 0.632; 10.34              |
> | ITI       | 0.709; 2.04; 5.06       | 0.920; 0.584; 16.33              |
> | Re-Control | 0.914; 1.99; 3.61       | 0.972; 0.656; 11.29              |
> | MAT-Steer | 0.697; 2.29; 3.86       | 0.912; 0.569; 21.66              |
> | **Ours**  | **0.686; 1.93; 7.23**   | **0.880; 0.560; 26.96**          |
>
>
> ### Relative Negative Representative Target Score
>
> | Method     | HelpSteer2 (Div; L1; SR) [3,3,3,2,2] | Code-UltraFeedback (Div; L1; SR) [2,2,2,2,2] |
> |-----------|--------------------------|----------------------------------|
> | Base      | 0.930; 2.79; N/A        | 0.963; 4.58; N/A                |
> | Prompting | 0.962; 2.78; 1.25       | 0.974; 4.57; 0.31               |
> | ITI       | 0.656; 2.83; 2.02       | 0.778; 4.62; 1.65               |
> | Re-Control | 0.888; 2.82; 2.49       | 0.917; 4.56; 0.72               |
> | MAT-Steer | 0.644; **2.72**; 2.73       | 0.651; 4.59; 1.65               |
> | **Ours**  | **0.602**; 2.83; **3.74**   | **0.605**; **2.89**; **19.14**           |
>
>
>
>
>
> > Consider incorporating discussions of recent multi-objective alignment literature to provide better context and positioning relative to the current state of the field.
>
> Rewards-in-Context [2] was already cited in the original manuscript (lines 684–686). Following your suggestion, we have added PARM [1], Controllable Preference Optimization (CPO) [3], and Preference Orchestrator [4] to the related-work section (Appendix B) and clarified their relationship to PRE-CONTROL (we note that Preference Orchestrator is also an ICLR 2026 submission and thus contemporaneous under the ICLR guidelines; while we were not required to compare to it in the initial submission, we have now done so in the revision).
>
> We appreciate being pointed to these papers and have incorporated them into the revision. Briefly, these methods study multi-objective alignment at test time by conditioning on **preference vectors or preference tokens**, typically via reward models or preference-aware adapters, to navigate trade-offs among multiple objectives. In contrast, PRE-CONTROL focuses on **precise, continuous attribute-intensity control** via **representation editing**, using a TD-trained value function over hidden states and a gradient-based intervention mechanism. Our approach is complementary to these lines of work: in principle, each attribute dimension in our target vector could correspond to an objective in those frameworks, while our method specifically targets fine-grained, continuous control over attribute strength.
>
>
> ---
> We are grateful for the constructive feedback provided by the reviewer. We have carefully addressed each of your concerns, including validation on the larger model and expanded discussion of recent multi-objective alignment literature. These additions, now incorporated into the revised manuscript with clear highlighting, demonstrate PRE-CONTROL's scalability and better position our work within the current research landscape. We believe these enhancements significantly strengthen the paper and hope our responses fully address your concerns, encouraging your support for acceptance.

---

> > ### Comment · Reviewer_HVF5 · 2025-11-25
> >
> > Thank you for the response. After considering the opinions of other reviewers, I have decided to maintain my score.

---

### Author Response · Authors · 2025-12-03
**Rebuttal Summary**

**To AC and SAC:**

We thank the AC/SAC for managing the review process and the reviewers for their constructive feedback

Due to the exceptional circumstances of this year’s discussion phase, reviewers were unable to respond to our rebuttal. However, we have addressed every concern raised, conducted all requested experiments, clarified misunderstandings by pointing to specific evidence in the original manuscript or providing new explanations, and incorporate major changes to the revised manuscript. We respectfully ask the AC to evaluate the paper based on these substantial efforts and the improvements they represent.

## Overall of Major Changes
We have made the following substantial revisions to address the reviewers' key concerns:
- **Scalability Verification**: Addressing Reveiwer-HVF5, VUE9, and bFK2, we conducted experiments on the Phi-4-14B model. Results in the expanded Table 1 confirm that PRE-CONTROL scales effectively, consistently achieving the highest success rate and diversity against all baselines.
- **Computational Cost Profiling**: In addition to our original statistics on value-function training and inference-time intervention, we expanded our computational cost analysis (Reviewer bFK2). Specifically, we compare PRE-CONTROL’s cost against all baselines and examine how cost scales with output-token length and batch size (Appendix F).
- **Hyperparameter Study**: We included a detailed hyperparameter sensitivity analysis (Reviewer bFK2) on step size, demonstrating our method's robustness across a wide range of step size values (Appendix H).
- **Clarified Novelty and Core Contribution**: Addressing Reviewer-VUE9 and SNhh, we sharpened the distinction between PRE-CONTROL and prior work (COLD, BOLT, PPLM) in Appendix B. We highlight our two core innovations: (1) a target-reaching objective (precise scalar intensity vs. simple reward maximization) and (2) An efficient intervention mechanism that uses a lightweight, TD-trained value function directly on hidden states, avoiding the expensive logit-space gradients of other methods.
- **Enhanced Related Work**: We have incorporated the suggested multi-objective alignment literature (Reviewer HVF5) in Section B.1, clarifying distinctions from prior work.
## Point-by-point Responses to Reviews
### Reviewer HVF5
- **Experiments on Larger-size Model**: We incorporated Phi-4-14B model experiment results in Table 1, validating our method's scalability with model size.
- **Multi-Objective Alignment Related Work**: We expanded our discussion of multi-objective related work (ARM, CPO, etc.) in Appendix B.
### Reviewer VUE9
- **Novelty vs. COLD/BOLT**: We clarified the fundamental differences in our optimization objective (target-reaching vs. reward-maximization) and mechanism (value-function steering vs. EBM search). We incorporate these in Related Work discussion (Appendix B).
- **Metric Clarification**: We added precise description of Success Rate in line 291-294.
- **Arbitrary Targets**: We clarified that our main targets were "representative" (based on data distribution, Fig. 5) and confirmed our method's robustness on a wide range of arbitrary targets in Appendix C.3 (Tables 14 & 15).
### Reviewer bFK2
- **Computational Overhead**: We provided a detailed computational cost analysis in Appendix F (Table 9), showing the overhead is a practical and modest trade-off for precise control.
- **Evaluation Setup**: We clarified that our research goal is to measure the precision of the control mechanism given a reward signal. The design choice of using ArmoRM is valid and appropriate given its exceptional performance as a human proxy.
- **Hyperparameter Sensitivity**: We incorporated a sensitivity analysis for intervention step size $\alpha$ in Appendix H (Figure 12), demonstrating our method's robustness.
### Reviewer SNhh
- **Novelty vs. PPLM**: We clarified our method's efficiency (gradients from a lightweight $V_{\phi}$ on hidden states vs. PPLM's expensive logit-space gradients) and objective (precise scalar targeting vs. binary property maximization).
- **Generation Quality Metric**: We evaluated our method using an additional generation quality metric, *coherence*, demonstrating that PRE-CONTROL does not trade off good generation quality for enhanced controllability.
- **Qualitative Example**: We enhanced the qualitative example in Appendix G to provide clearer insight into our method, specifically by showing cases where we control only a single attribute.
- **Comparison with SFT**: We compared our method with RiC, a SFT multi-objective alignment method. Results support that PRE-CONTROL achieves better Hypervolume and success rate with significant less computational overhead.

**Conclusion:** Over the past month, we have put forth our best efforts to strengthen the paper and address each concern raised by the reviewers. We trust the AC will consider these improvements and the specific circumstances of the discussion phase when making a final recommendation.

---

### Note · Program_Chairs · 2026-01-17
**Submission Desk Rejected by Program Chairs**

The following references in this submission do not refer to real documents and/or have major errors in bibliographic information:

 Jacob Eisenstein, Jure Leskovec, Emily M. Bender, and Chris Callison-Burch. Preference-based learning for user-centered natural language generation. ACM Transactions on Interactive Intelligent Systems, 13 (1):1-30, 2023.